# No RL, No Simulation: Learning to Navigate without Navigating

**Meera Hahn** *
Georgia Institute of Technology

**Devendra Chaplot**
Facebook AI Research

**Shubham Tulsiani**
Facebook AI Research

**Mustafa Mukadam**
Facebook AI Research

**James M. Rehg**
Georgia Institute of Technology

**Abhinav Gupta**
Facebook AI Research

## Abstract

Most prior methods for learning navigation policies require access to simulation environments, as they need online policy interaction and rely on ground-truth maps for rewards. However, building simulators is expensive (requires manual effort for each and every scene) and creates challenges in transferring learned policies to robotic platforms in the real-world, due to the sim-to-real domain gap. In this paper, we pose a simple question: Do we really need active interaction, ground-truth maps or even reinforcement-learning (RL) in order to solve the image-goal navigation task? We propose a self-supervised approach to learn to navigate from only passive videos of roaming. Our approach, No RL, No Simulator (NRNS), is simple and scalable, yet highly effective. NRNS outperforms RL-based formulations by a significant margin. We present NRNS as a strong baseline for any future image-based navigation tasks that use RL or Simulation.

## 1 Introduction

In recent years, we have seen significant advances in learning-based approaches for indoor navigation [1, 2]. Impressive performance gains have been obtained for a range of tasks, from non-semantic point-goal navigation [3] to semantic tasks such as image-goal [4] and object-goal navigation [5, 6], via methods that use reinforcement learning (RL). The effectiveness of RL for these tasks can be attributed in part to the emergence of powerful new simulators such as Habitat [7], Matterport [8] and AI2Thor [9]. These simulators have helped scale learning to billions of frames by providing large-scale active interaction data and ground-truth maps for designing reward functions. But do we actually need simulation and RL to learn to navigate? Is there an alternative way to formulate the navigation problem, such that no ground-truth maps or active interaction are required? These are valuable questions to explore because learning navigation in simulation constrains the approach to a limited set of environments, since the creation of 3D assets remains costly and time-consuming.

In this paper, we propose a self-supervised approach to learning how to navigate from passive egocentric videos. Our novel method is simple and scalable (no simulator required for training), and at the same time highly effective, as it outperforms RL-based formulations by a significant margin. To introduce our approach, let us first examine the role of RL and simulation in standard navigation learning methods. In the standard RL formulation, an agent gets a reward upon reaching the goal, followed by a credit assignment stage to determine the most useful state-action pairs. But do we actually need the *reinforce* function for action credit assignment? Going a step further, do we even need to learn a policy explicitly? In navigation, we argue that the state space itself is highly structured via a distance function, and the structure itself could be leveraged for credit assignment. Simply put, states that help reduce the distance to the goal are better – and therefore predicted distance can be

---

* Correspondence: meerahahn@gatech.edu

35th Conference on Neural Information Processing Systems (NeurIPS 2021).

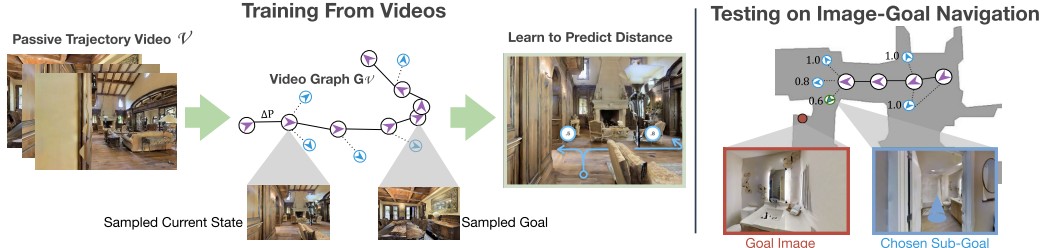

Figure 1: **Left**: Using passive videos we learn to predict distances for navigation. Our distance function learns the priors of the layouts of indoor buildings to estimate distances to goal location. **Right**: Image-Goal Navigation Task [10]. Our model uses distance function to predict distances of unexplored nodes and uses greedy policy to choose the shortest-distance node.

used either as the value function or as a proxy for it. In fact, RL formulations frequently use 'distance reduced to goal' in reward shaping. The key property of our approach is that we learn a generalizable distance estimator *directly* from passive videos, and as a result we do not require any interaction. We demonstrate that an effective distance estimator can be learned directly from visual trajectories, without the need for an RL policy to map visual observations to the action space, thereby obviating the need for extensive interaction in a simulator and hand-designed rewards. However passive videos do not provide learning opportunities for obstacle avoidance since they rarely, if ever, consist of cameras bumping into walls. We forgo the need for active interaction to reason about collisions as we show that obstacle avoidance is only required locally and simple depth maps are sufficient to prune invalid actions and locations for navigation. More broadly, our approach can be considered as closely related to model-based control, which is an alternative paradigm to RL-based policy learning, with the key insight that components of the model and cost functions can be learned from passive data.

**No RL, No Simulator Approach (NRNS):** Our NRNS algorithm can be described as follows. During training we learn two functions from passive videos: (a) a *geodesic distance estimator*: given the state features and goal image, this function predicts the geodesic distance of the unexplored frontiers of the graph to the goal location. This enables a greedy policy in which we select the node with the least distance; (b) *target prediction model*: Given the goal image and the image from the agent's current location, this function predicts if the goal is within sight and can be reached without collisions, along with the exact location of the goal. The key is that both the distance model and the target prediction model can be learned from passive RGBD videos, with SLAM used to estimate relative poses. We believe our simple NRNS approach should act as a strong baseline for any future approaches that use RL and simulation. We show that NRNS outperforms end-to-end RL, even when RL is trained using 5x more data and 10x more compute. Furthermore, unlike RL methods which need to be trained in simulation because they require substantial numbers of interactions, NRNS can be trained directly on real-world videos alone, and therefore does not suffer from the sim-to-real gap.

## 2   Related Work

**Navigation in simulators.** Navigation tasks largely fall into two main categories [1], ones in which a goal location is known [11, 12, 13] and limited exploration is required, and ones in which the goal location is not known and efficient exploration is necessary. In the second category, tasks range from finding the location of specific objects [5], rooms [14], or images [15], to the task of exploration itself [2]. The majority of current work [12, 15, 16, 3] leverages simulators [7] and extensive interaction to learn end-to-end models for these tasks. In contrast, our work shows that the semantic cues needed for exploration-based navigation tasks can be learned directly from video trajectories.

**Navigation using passive data.** Several prior works have tackled the navigation task when there is some passive experience available in the test environment [17, 18, 19, 20, 21, 22]. A more limited number of works train navigation policies without simulation environments [19, 22, 23]. Unlike these works, we tackle the task of image goal navigation in unseen environments where there is no experience available to the agent in the test environments during training [19, 22] and no map knowledge is given about the environments [23]. Two works adopting similar requirements are Chaplot et al. [4] and Chang et al. [24]. Most closely related is Neural Topological SLAM (NTS) [4], which builds a topological map and estimates distance to the target image using a learned function. But NTS requires access to panoramic observations and ground-truth maps for training the distance

function, making it much less scalable than our approach which can be trained with just video data with arbitrary field of view. Chang et al. [24] adopt a similar approach to NTS for object goal navigation, while also incorporating video data from YouTube for learning a Q function. A key difference is that our method learns a episodic distance function, utilizing all past observations to estimate distances to the target image. In comparison, Chaplot et al. [4] and Chang et al. [24] use a memory-less distance function operating only on the agent's current observation.

Prior work work on LfD [25, 26, 27] requires: (a) access to actions (e.g. teleoperated in simulator); and (b) optimal human demonstrations. The goal in LfD is to learn policies that match given demonstrations by either mimicking the actions or formulating rewards using the demonstrations. In contrast, NRNS does neither, since our passive training data does always contain optimal navigation trajectories and NRNS does not compute any rewards as it does not require RL or credit assignment. We make the same distinction between our work and offline RL [28, 29, 30], which does not utilize online data gathering but still learns a policy using rewards from observed trajectories. In contrast, we do not use any rewards or learn a policy. Instead, our approach aims to learn a distance-to-goal model and uses a greedy policy based on the distance predictions. From this perspective, NRNS is performing model-based learning.

**Map-Based Navigation.** There are multiple spatial representations which can be leveraged in solving navigation tasks. Metric maps, which maintain precise information on the occupied space in an environment, are commonly used for visual navigation [2, 10, 6]. However, metric maps suffer from issues of scalability and consistency under sensor noise. As a result, topological maps have recently gained traction [4, 21, 18, 24, 31] as a means to combat these issues. A significant difference from our approach is that in these prior works a topological map is given at the beginning of the navigation task, and is not created or changed during navigation. Specifically, Savinov et al. [18] create the map from a given 5 minute video of the test environment and Chen et al. [21] assume access to a ground truth map. In our image-goal navigation set up, the agent is given no information about the test environment. Chaplot et al. [4] do not require experience in the test environment and build topological maps at test time, but still requires access to a simulator for computing shortest-path distances between pairs of images. Additionally, topological maps for robotic navigation draw inspiration from both animal and human psychology. The cognitive map hypothesis proposes that the brain builds coarse internal spatial representations of environments [32, 33]. Multiple works argue that this internal representation relies on landmarks [34, 35], making human cognitive maps more similar to topological maps as opposed to metric maps.

**Graph Neural Networks.** Graph Neural Networks (GNN) are specifically used for modeling relational data in a non-Euclidean space. We employ a GNN for estimating distance-to-goal over the agent's exploration frontier. Specifically we employ an augmented Graph Attention Network [36] which allows for a weighted aggregation of neighboring node information. Graph Networks are rarely-used for the task of topological map-based visual navigation. Savinov et al. [18] use a GNN for the sub-task of localizing the agent in a ground truth topological map. We believe we are the first to leverage graph neural networks in a visual navigation task.

## 3  Image Goal Navigation using Topological Graphs

We propose No RL, No Simulator "NRNS", a hierarchical modular approach to image-goal navigation that consists of: a) high-level modules for maintaining a topological map and using visual and semantic reasoning to predict sub-goals, and b) heuristic low-level modules that use depth data to select low level navigation actions to reach sub-goals and determine geometrically-explorable areas. We first describe NRNS in detail and then show in Sec. 4 that the high-level modules can be trained without using any simulation, interaction or even ground-truth scans – i.e. that passive video data alone is sufficient to learn the semantic and visual reasoning models needed for navigation.

### 3.1  Formulation and Representation

**Task Definition.** We tackle the task of image-goal navigation, where an agent is placed in a novel environment, and is tasked with navigating to an (unknown) goal position that is specified using an image taken from that position, as shown in Fig. 2. More formally, an episode begins with an agent receiving an RGB observation ($I_G$) corresponding to the goal position. At each time step, $t$, of the episode the agent receives a set of observations $s_t$, and must take a navigation action $a_t$. The agents state observations, $s_t$, are defined as a narrow field of view RGBD image, $I_t$, and egocentric pose

estimate, $P_t$. The agent must execute navigational actions to reach goal within a maximum number of steps.

**Topological Map Representation.** The NRNS agent maintains a topological map in the form of a graph $G(N, E)$, defined by nodes $N$ and edges $E$, which provides a sparse representation of the environment. Concretely, a node $n_i \in N$ is associated with a pose $p_i$, defined by location and orientation. Each node $n_i$ can either be 'explored' *i.e.* the agent has previously visited the pose and obtained a corresponding RGBD image $I_i$, or 'unexplored' *e.g.* unvisited positions at the exploration frontier which may be visited in the future. Each edge $e \in E$ connects a pair of adjacent nodes $n_i$ and $n_j$. Nodes are deemed adjacent only if a short and 'simple' path exists between the two nodes, as further detailed in Sec. 3.3. Each edge between adjacent nodes is then associated with the attribute $\Delta P$ – the relative pose between the two nodes.

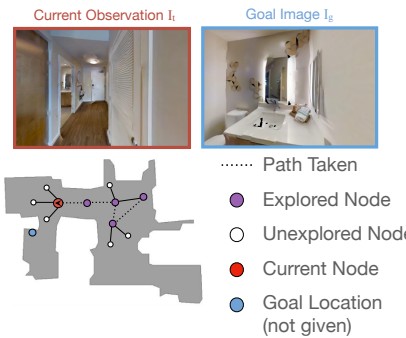

Figure 2: Image-Goal navigation task using a topological graph.

## 3.2 Global Policy via Distance Prediction

Given a representation of the environment as a topological graph, our global policy is tasked with identifying the next 'unexplored' node that the agent should visit, and the low-level policy ($\mathcal{G}_{LP}$) is then responsible for executing the precise actions. Intuitively, we want our agent to select as the next node the one that minimizes the total distance to goal. The global policy's inference task can thus be reduced to predicting distances from nodes in our graph to the goal location. To this end, our approach leverages a distance prediction network ($\mathcal{G}_D$) which operates on top of $G(N, E)$ to predict distance-to-goal for each unexplored node $n_{ue} \in G$. Our global policy then simply selects the node with least total distance to goal which is defined as: distance to the unexplored node from the agent's current position, plus the predicted distance from the unexplored node to the goal.

The input to the distance prediction network $\mathcal{G}_D$ is the current topological graph $G(N, E)_t$ and $I_G$. While the explored nodes have an image associated with them, the unexplored nodes naturally do not. To allow prediction in this setup, we use a Graph Convolutional Network (GCN) architecture to first induce visual features for $n_{ue} \in G$, and then predict the distance to goal using an MLP.

As illustrated in Fig. 3, the network first encodes the RGB images at each explored node ($n_i$) using a ResNet18 [37] to obtain feature vector $\mathbf{h_i} \in \mathbb{R}^{512}$. Each edge $e_{i,j}$ is further represented by a feature vector $\mathbf{u_{i,j}}$, which is the flattened pose transformation matrix ($^{\mathbf{i}}\mathbf{K_j} \in \mathbb{R}^{4\times4}$). The adjacency matrix $\mathbf{A_t}$, $\mathbf{u_{i,j}}$, and $\mathbf{h_i}$ are passed through a GCN comprising of two graph attention (GAT) layers [36] with intermediate nonlinearities. Note that we extend the graph attention layer architecture to additionally use edge features $\mathbf{u_{i,j}}$ when computing attention coefficients. The predicted visual features for unexplored nodes are then finally used to compute predicted distance-to-goal $d_i$ from each node to $I_G$ using a simple MLP.

To select the most 'promising' $n_{ue}$ to explore, the distance from the agent's current location $n_t$ also needs to be accounted for. For $n_{ue}, n_t \in G$, the 'travel cost' is added to $d_i$, calculated using shortest path on G from $n_{ue} \rightarrow n_t$. Our global policy then selects the unexplored node with the minimum total distance score as the next sub-goal.

## 3.3 Local Navigation and Graph Expansion

The NRNS global policy selects the sub-goal that the agent should pursue, and the precise low-level actions to reach the sub-goal are executed by a heuristic local policy. After the local policy finishes execution, the NRNS agent updates the graph to include the new observations and expands the graph with unexplored nodes at the exploration frontier.

**Local Policy.** The NRNS local policy, denoted as $\mathcal{G}_{LP}$, receives a target position, defined by distance and angle ($\rho_i$, $\phi_i$) with respect to the agent's current location. When $\mathcal{G}_D$ outputs a sub-goal node, $\rho_i$, $\phi_i$ are calculated from the current position and passed to $\mathcal{G}_{LP}$. Low level navigation actions are selected and executed using a simplistic point navigation model based on the agent's egocentric RGBD observations and (noisy) pose estimation. To navigate towards its sub-goal, the agent builds

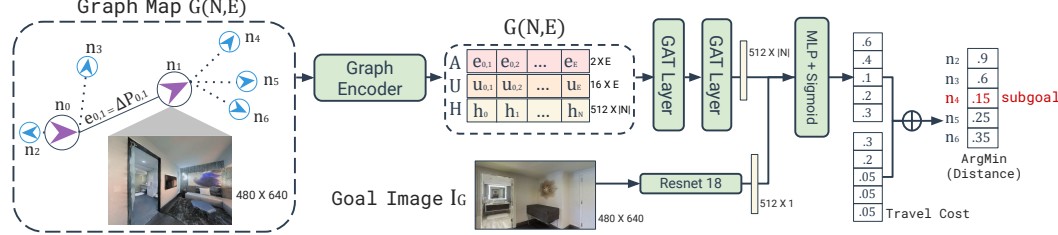

Figure 3: The Global Policy and $\mathcal{G}_D$ architecture used to model distance-to-goal prediction. $\mathcal{G}_D$ employs a Resnet18 encoder, Graph Attention layers, and a multi-layer perceptron with a sigmoid.

and maintains a local metric map using the noisy pose estimator and depth input. This effectively allows it to reach local goals and avoid obstacles. The local metric maps are discarded upon reaching the sub-goal, as they are based on a noisy pose estimator. This policy is adapted from [10] and is also used for Image-Goal Navigation in [4].

**Explorable Area Prediction for Graph Expansion.** We incorporate a 'graph expansion' step after the agent reaches a sub-goal via $\mathcal{G}_{LP}$ and before the agent selects a new sub-goal. First, the agent updates $G(N, E)$ to record the current location as an explored node and store the associated RGBD observation. Second, the agent creates additional 'unexplored' nodes, $n_{ue}$, adjacent to the current node, $n_t$ based on whether the corresponding directions are deemed 'explorable'. We use a explorable area prediction module, $\mathcal{G}_{EA}$, to determine which adjacent areas to the current location are geometrically explorable. This is untrained, heuristic module takes the egocentric depth image $I_{t_{depth}}$ and tests 9 angles in the direction $\theta$ from the current position and returns the angles $\theta$ that are not blocked by obstacles within a depth of 3 meters. The NRNS agent tests $\theta$ at $[0, \pm 15, \pm 30, \pm 45, \pm 60]$, these angles are chosen based on the agent's turn radius of 15° and 120° FOV. For all $\theta$ determined to be explorable, the agent updates $G(N, E)$ by adding an 'unexplored' node at position $\rho = 1m$ and $\phi = \theta$ relative to the node the agent is at, with a corresponding edge linking the current node to the new node. If a node already exists in one of the explorable areas at a nearby position, only an edge is added to $G(N, E)$ and not a new node.

## 3.4 Putting it Together

**Stopping Criterion.** The above described modules ($\mathcal{G}_D$, $\mathcal{G}_{LP}$, $\mathcal{G}_{EA}$) allow the NRNS agent to incrementally explore its environment while making progress towards the location of the target image. To allow the agent to terminate navigating when it is near the goal, we additionally learn a target prediction network $\mathcal{G}_T$ that provides a stopping criterion. Given $I_G$ and current image $I_t$, $\mathcal{G}_T$ is simple MLP that predicts: a) a probability $\beta_s \in [0, 1]$ indicating whether the goal is within sight, and if so, b) the relative position (distance, direction) of the goal $\rho_g$, $\phi_g$.

**Algorithm.** We outline our navigation algorithm in Fig. 4. An example result on the image-goal navigation task is shown in Fig. 6. Among the modules used, the local policy module $\mathcal{G}_{LP}$ and the explorable area module $\mathcal{G}_{EA}$ do not require any learning. As we show in Sec. 4, $\mathcal{G}_D$ and $\mathcal{G}_T$ can both be learned using only passive data.

---

**Algorithm 1:** NRNS Image Navigation

```
// initialize graph
n₀ = (P_{t=0}, I_{t=0}); e₀ = (n₀, n₀);
N = (n₀); E = (e₀);
G = (N, E);
// loop until reached goal or max steps
while steps taken < max steps do
    n_{i+1}, ..., n_{i+k} = G_EA(I_t); // determine valid explorable areas
    N += n_{i+1}, ..., n_{i+k}; E += e_{t,i+1}, ..., e_{t,i+k}; // add unexplored nodes & edges
    n_{sg}, (ρ_{sg}, φ_{sg}) = argmin(G_D(G, I_G) + TravelCost(G, n_t)); // select sub-goal
    I_{t+1}, P_{t+1} = G_LP(ρ_{sg}, φ_{sg}); // navigate to sub-goal
    n_{sg} = (P_{t+1}, I_{t+1}) // update graph with observations
    β_s, (ρ_g, φ_g) = G_T(I_{t+1}, I_G); // stopping criterion
    if β_s > .5 then
        G_LP(ρ_g, φ_g); // navigate to target
        break;
    end
end
```

---

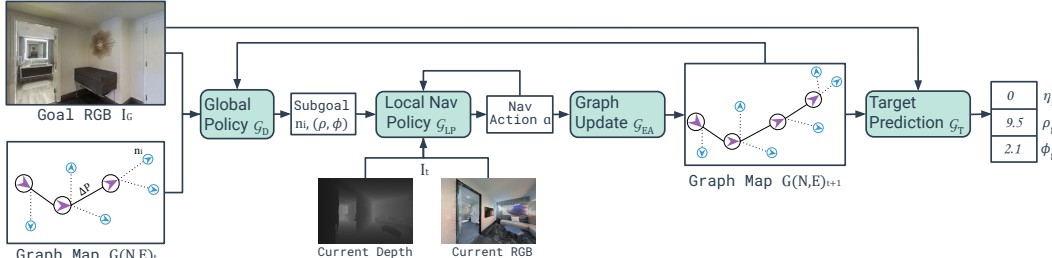

Figure 4: The hierarchical modular NRNS approach to the Image-Goal Navigation task, for a single time step in the episode. The global policy $\mathcal{G}_D$ selects an unexplored node $n_i$ as a sub-goal. The sub-goal position $(\rho_i, \phi_i)$ is passed to the local navigation policy $\mathcal{G}_{LP}$ which takes in the current RGBD observations and outputs low level actions until the agent reaches $n_i$. The graph is then updated with the current observations $I_{t+1}$ and new unexplored nodes and edges generated by $\mathcal{G}_{EA}$.

## 4   Learning from Passive Data

The learned high-level policies of NRNS are the Distance Network $\mathcal{G}_D$ and Target Prediction Network $\mathcal{G}_T$. A key contribution of our work is showing that these functions can be learned from passive data alone. This eliminates the need for online interaction and ground-truth maps, allowing us to train the NRNS algorithm without using RL or simulation.

**Learning Distance Prediction.** First, we describe how to learn the function $\mathcal{G}_D$. Given a topological graph (consisting of both explored and unexplored nodes) and a goal image as input, $\mathcal{G}_D$ predicts the geodesic distance from all unexplored nodes to the location of the goal image. Our training data therefore consists of triplets of the form $(G, I_G, D_U)$, where $G$ is a topological graph, $I_G$ is a goal image, and $D_U$ is the ground-truth distances from a set of unexplored nodes to the goal location (we use L2-Loss to train the distance function).

We generate training graphs using passive videos in a two-step process. In the first step, we create a topological graph, $G_{\mathcal{V}_i}$, for every video $\mathcal{V}_i$ in the passive dataset. Each graph contains both explored and unexplored nodes. We approximate distance to unexplored nodes using the geodesic distance along the trajectory. In the second step, we uniformly sample sub-graphs and goal locations over each video's topological graph.

**Step 1: Video to Topological Graph.** First, we generate a step-wise trajectory graph in which each frame is a node with odometry information. We then process this step-wise graph into a topological graph using affinity clustering [38]. We use the visual features from each frame concatenated with odometry information as the feature vector for clustering. Visual features for each frame are extracted via a Places365 [39] pretrained Resnet18. Each cluster represents a single node in the topological graph $G_{\mathcal{V}_i}$ and the stepwise visual features of frames in the cluster are average pooled. The topological graphs are then expanded to have unexplored nodes, used in training the distance function. These are created by applying $\mathcal{G}_{EA}$ to the RGBD of each node centroid in $G_{\mathcal{V}_i}$. Fig. 5 illustrates the process by which an example video is converted into a topological graph for training data.

**Step 2: Sampling training datapoints.** Individual data instances are selected via uniform sampling without replacement. This means selecting a random node in $G_{\mathcal{V}}$ as the goal location and a random sub-graph of $G_{\mathcal{V}}$ as the observed trajectory and current location. Distance along the trajectory is used as the ground truth distance between nodes in the sub-graph and the goal image: these distance labels are used to train the network $\mathcal{G}_D$. Due to non-optimal long term paths of the video trajectories, distances may overestimate distance to the goal. In other words while each node along the generated trajectory graphs are legitimate paths to the goal, shorter paths may exist that are not covered in the training trajectory graph, making this a challenging problem. In total, $\mathcal{G}_D$ is trained using ~1k training instances per scan for both Gibson and MP3D.

**Learning Target Direction Prediction** The $\mathcal{G}_T$ prediction network receives a goal image and the current node image as input, and predicts whether the goal is within sight of the current image. To gather training instances for $\mathcal{G}_T$, we use the RGB frame at a randomly selected node as the goal image. We make a simplifying assumption that any adjacent pair of nodes (similar features and odometry) in the topological graph are positive examples and any other pair of nodes are negative examples.

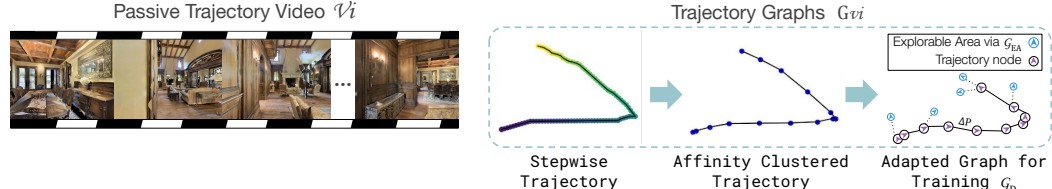

Figure 5: Example from the passive video dataset. Frames of a video trajectory $\mathcal{V}_i$ are shown on the left. The stepwise trajectory is then turned into a trajectory graph $G_{\mathcal{V}_i}$ via affinity clustering [38] of node image and pose features. $G_{\mathcal{V}_i}$ is adapted to train the Global Policy $\mathcal{G}_D$ and target direction prediction $\mathcal{G}_E A$. An example of an adapted $G_{\mathcal{V}_i}$ for training is shown on the left.

## 5 Experiments

**Image-Goal Navigation Task Setup.** At the beginning of each episode, the agent is placed in an unseen environment. The agent receives observations from the current state, a 3 x 1 odometry pose reading and RGBD image, and the RGB goal image. Observation and goal images are 120°FOV and of size 480 x 640. An episode is considered a success if the agent is able to reach within $1m$ of the goal location and do so within a maximum episode length of $500$ steps. Each episode is also evaluated by the efficiency of the navigational path from start to goal, which is quantitatively measured by Success weighted by inverse Path Length (SPL) [1]. Note, the narrow field of the agent in this task definition differs from past works which use panoramic views [4]. The decision to use a narrow field is based on our method of training only on passive data. Current passive video datasets of indoor trajectories such as YouTube Tours [24], RealEstate10k [40] and our NRNS passive video dataset, do not contain panoramas.

**Action Space.** The agent's action space contains four actions: `forward` by .25m, `rotate left` by 15°, `rotate right` by 15°, and `stop`. In our experiments, we consider two cases for pose estimation and action transition. In the first condition the agent has access to ground truth pose and the navigation actions are deterministic. In the second condition, noise is added to the pose estimation and the actuation of the agent. We utilize the realistic pose and actuation noise models from [10], which are similarly used in [4]. The actuation noise adds stochastic rotational and translations transitions to the agent's navigational actions.

**Training Data.** Our key contribution is the ability to learn navigation agents from passive data. In theory, our approach can be trained from any passive data source, and we test this in Sec. 5.2 using RealEstate10K [40]. However, since RL-based baselines are trained in the Habitat Simulator [7], we generate our NRNS dataset, of egocentric trajectory videos, using the same Habitat training scenes to provide direct comparison and isolate domain gap issues. Specifically, the trajectory videos are created as follows. A set of 2 - 4 points are randomly selected from the environment using uniform sampling. A video is then generated of the concatenated RGBD frames of the shortest path between consecutive points. Note that the complete video trajectory (from first source to final target) is not step-wise optimal. Frames in the videos are of size 480 X 640 and have a FOV 120°and each frame is associated with a 3 x 1 odometry pose reading. In the noisy setting discussed in Sec. 5, sensor and actuation noise is injected into training trajectories. We create 19K, 43K video trajectories, containing 1, 2.5 million frames respectively, on the Gibson and MP3D datasets. We then use this data to train the NRNS modules, as described in Sec. 4.

**Test Environments.** We evaluate our approach on the task of image-goal navigation. For testing, we use the Habitat Simulator [7]. We evaluate on the standard test-split for both the Gibson [41] and Matterport3D (MP3D) [42] datasets. For MP3D, we evaluate on 18 environments and for Gibson, we evaluate on 14 environments.

**Baselines.** We consider a number of baselines to contextualize our Image-Goal Navigation results:

- **BC w/ ResNet + GRU.** Behavioral Cloning (BC) policy where $I_t$ and $I_G$ are encoded using a pretrained ResNet-18. Both image encodings and the previous action $a_{t-1}$ are passed through a two layer Gated Recurrent Unit (GRU) with softmax, which outputs the next action $a_t$.
- **BC w/ ResNet + Metric Map.** BC policy: $I_t$ and $I_G$ are encoded with a ResNet, same as the above policy. This policy keeps a metric map built from the depth images. The metric map is encoded with a linear layer. The metric map encoding and encodings of $I_t$ and $I_G$ are concatenated and passed into an MLP with softmax, which outputs the next navigational action $a_t$.

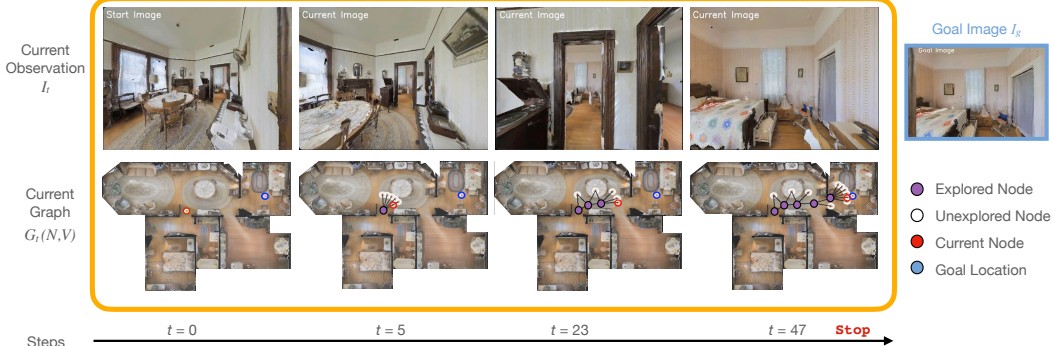

Figure 6: Example of an Image-Goal Navigation episode on MP3D. Shows the agent's observations and internal topological graph at different time steps.

  – **End to End RL with DDPPO.** An agent is trained end to end with proximal policy optimization [13] in the Habitat Simulator [7] for the image-goal navigation task.

We use and adapt code for baseline algorithms from Chaplot *et al.* [4]. However, as our setup uses narrow-view cameras instead of panoramas, these adapted baselines perform worse compared to their previously reported performance. This difference in setup also makes direct comparison with [4] infeasible, as they critically rely on panoramic views for localization. The baseline behavioral cloning policies are also trained only using the passive dataset described in Sec. 4. This makes the BC baseline policies directly comparable to the NRNS model.

The end-to-end RL policy is trained using a Habitat implementation of DDPPO [13]. For training we use 8 GPUs and 16 processes per GPU. We first train with 1k episodes per house, which is identical to how NRNS is trained, and we train for 10M steps. To test the scalability of the RL method, we additionally train a model using 5k training episodes per environment (providing the RL agent 5x more data than our NRNS agent) and report the performance of the agent trained on these episodes at 50M steps (5x more compute than NRNS) and 100M steps (10x more compute than NRNS). Training the $\mathcal{G}_D$ model takes $\sim$20 epochs, requiring $\sim$8 hours on a single GPU. Training the $\mathcal{G}_T$ model takes $\sim$10 epochs, requiring $\sim$4 hours on a single GPU.

**Episode Settings.** To provide an in-depth understanding of the successes and limitations of our approach, we sub-divide test episodes into two categories: 'straight' and 'curved'. In 'straight' episodes, the ratio of shortest path geodesic-distance to euclidean-distance between the start and goal locations is $< 1.2$ and rotational difference between the orientation of the start position and goal image is $< 45°$. All other start-goal location pairs are labeled as 'curved' episodes. We make this distinction due to the nature of the narrow field of view of our agent, which strongly affects performance on curved episodes, since the agent must learn to turn both as part of navigating and part of seeking new information about the target location. Also, while a greedy policy being successful on 'straight' episodes might be expected, a competitive performance on even 'curved' episodes will highlight how effective our simple model and policy is. We further subdivide each of these 2 categories into 3 sub-categories of difficulty: 'easy', 'medium' and 'hard'. Difficulty is determined by the length of the shortest path between the start and goal locations. Following [4], the 'easy', 'medium' and 'hard' settings are $(1.5 - 3m)$, $(3 - 5m)$, and $(5 - 10m)$ respectively. To generate test episodes we uniformly sample the test scene for start-goal location pairs, to create approximately 1000 episodes per setting.

## 5.1 Results

Tables 1, 2 show the performance of our NRNS model and relevant baselines on the test splits of the Gibson and Matterport datasets. In 6, we visualize an episode of the NRNS agent as it navigates to the goal-image.

**NRNS outperforms baselines.** Our NRNS algorithm outperforms the BC and end to end RL policies in terms of Success and SPL @ 1m on both datasets. NRNS improves upon the best offline baseline, a Behavioral Cloning (BC) policy with a ResNet and GRU, across splits of Gibson by an absolute 20+% on Straight episodes and 10+% on Curved episodes. We find that a BC policy using a GRU for memory outperforms using only a metric map. We attribute this to a spatial memory (metric map) being less informative for agent exploration than a memory of the visual features and previous

steps (GRU). We observe that an end-to-end RL policy trained for 10M steps with 10k epsiodes per house, in simulation performs much weaker than all baselines. With increased data and steps RL baselines unsurprisingly increase in performance, however we find with 5X more data and 10x more compute RL baselines still are outperformed by NRNS. The low performance of behavioral cloning and RL methods for image-goal navigation is unsurprising [4, 15]. This demonstrates the difficulty of learning rewards on low level actions instead of value learning on possible exploration directions, exacerbating the difficulty of exploration in image-goal navigation. Adding to the challenges of the task, all policies must learn the stop action. Previous works [4] have found that adding oracle stopping, to a target-driven RL agent, leads to large gains in performance on image-goal navigation. The limitations of all approaches are seen on the 'hard' and 'curved' episode settings, showing the overall difficulty of the exploration problem and the challenge of using a narrow field of view.

**NRNS is robust to noise.** Even with the injection of sensor and actuation noise [10], in both the passive training data and test episodes, NRNS maintains superior or near comparable performance to all baselines. In fact, we find that the addition of noise leads to only an absolute drop in success between .8-8% on Gibson [41] and 1-5% on MP3D [42]. An interesting observation is small increase in performance (w/noise) for the hard-case. We believe this is because gt-distance for hard cases are more error prone and noise during training provides regularization.

Table 1: Comparison of our model (NRNS) with baselines on Image-Goal Navigation on **Gibson**[41]. We report average Success and Success weighted by inverse Path Length (SPL) @ $1m$. Noise refers to injection of sensor & actuation noise into the train videos and test episodes. * denotes using simulator.

| Path Type | Model | Easy | | Medium | | Hard | |
|---|---|---|---|---|---|---|---|
| | | Succ ↑ | SPL ↑ | Succ ↑ | SPL ↑ | Succ ↑ | SPL ↑ |
| Straight | RL (10M steps) * [13] | 10.50 | 6.70 | 18.10 | 16.17 | 11.79 | 10.85 |
| | RL (extra data + 50M steps)* [13] | 36.30 | 34.93 | 35.70 | 33.98 | 5.94 | 6.33 |
| | RL (extra data + 100M steps)* [13] | 43.20 | 38.54 | 36.40 | 34.89 | 7.44 | 7.20 |
| | BC w/ ResNet + Metric Map | 24.80 | 23.94 | 11.50 | 11.28 | 1.36 | 1.26 |
| | BC w/ ResNet + GRU | 34.90 | 33.43 | 17.60 | 17.05 | 6.08 | 5.93 |
| | NRNS w/ noise | 64.10 | 55.43 | 47.90 | 39.54 | **25.19** | 18.09 |
| | NRNS w/out noise | **68.00** | **61.62** | **49.10** | **44.56** | 23.82 | **18.28** |
| Curved | RL (10M steps)* [13] | 7.90 | 3.27 | 9.50 | 7.11 | 5.50 | 4.72 |
| | RL (extra data + 50M steps)* [13] | 18.10 | 15.42 | 16.30 | 14.46 | 2.60 | 2.23 |
| | RL (extra data + 100M steps)* [13] | 22.20 | 16.51 | 20.70 | **18.52** | 4.20 | 3.71 |
| | BC w/ ResNet + Metric Map | 3.10 | 2.53 | 0.80 | 0.71 | 0.20 | 0.16 |
| | BC w/ ResNet + GRU | 3.60 | 2.86 | 1.10 | 0.91 | 0.50 | 0.36 |
| | NRNS w/ noise | 27.30 | 10.55 | 23.10 | 10.35 | 10.50 | 5.61 |
| | NRNS w/out noise | **35.50** | **18.38** | **23.90** | 12.08 | **12.50** | **6.84** |

Table 2: Comparison of our model (NRNS) with baselines on Image-Goal Navigation on **MP3D**[42].

| Path Type | Model | Easy | | Medium | | Hard | |
|---|---|---|---|---|---|---|---|
| | | Succ ↑ | SPL ↑ | Succ ↑ | SPL ↑ | Succ ↑ | SPL ↑ |
| Straight | RL (10M steps)* [13] | 7.50 | 4.00 | 3.50 | 1.73 | 1.00 | 0.55 |
| | RL (extra data + 50M steps)* [13] | 34.50 | 30.08 | 35.70 | 33.80 | 10.40 | 10.07 |
| | RL (extra data + 100M steps)* [13] | 36.40 | 30.84 | 33.80 | 31.42 | 12.00 | 11.56 |
| | BC w/ ResNet + Metric Map | 25.80 | 24.82 | 11.30 | 10.65 | 3.00 | 2.93 |
| | BC w/ ResNet + GRU | 30.20 | 29.57 | 12.70 | 12.48 | 4.40 | 4.34 |
| | NRNS w/ noise | 63.80 | 53.12 | 36.20 | 26.92 | **24.10** | 16.93 |
| | NRNS w/out noise | **64.70** | **58.23** | **39.70** | **32.74** | 22.30 | **17.33** |
| Curved | RL (10M steps)* [13] | 4.90 | 1.78 | 3.20 | 1.37 | 1.10 | 0.46 |
| | RL (extra data + 50M steps)* [13] | 15.70 | 11.34 | 10.70 | 9.03 | 3.90 | 3.57 |
| | RL (extra data + 100M steps)* [13] | 17.90 | **13.24** | 15.00 | **12.17** | 5.90 | 4.87 |
| | BC w/ ResNet + Metric Map | 4.90 | 4.23 | 1.40 | 1.29 | 0.40 | 0.34 |
| | BC w/ ResNet + GRU | 3.10 | 2.61 | 0.80 | 0.77 | 0.10 | 0.02 |
| | NRNS w/ noise | 21.40 | 8.19 | 15.40 | 6.83 | **10.0** | 4.86 |
| | NRNS w/out noise | **23.70** | 12.68 | **16.20** | 8.34 | 9.10 | **5.14** |

**NRNS Module Ablations.** Tab. 3 reports detailed ablations of NRNS on the Gibson dataset (see appendix for ablation results on MP3D). We ablate the NRNS approach by testing each module individually. In the ablation experiments, we replace the module output with the ground truth labels or numbers in order to evaluate the affect of each module on the performance of the overall approach. For simplicity, all ablations are trained and tested without sensor or actuation noise. Unsurprisingly, we find that the Global Policy, $\mathcal{G}_D$, has a large affect on performance (Row 4 and 8). We find that the largest affects are seen in the 'hard' and 'curved' test episodes. This is unsurprising because as the distance to the goal increases, the path increases in complexity and the search space of $\mathcal{G}_D$ increases.

Table 3: Ablations of NRNS with baselines on Image-Goal Navigation on **Gibson** [41]. We report average Success and Success weighted by inverse Path Length (SPL) @ $1m$. ✗ denotes a module being replaced by the ground truth labels and a ✓ denotes the NRNS module being used.

| Path Type | NRNS Ablation | | | Easy | | Medium | | Hard | |
| | $\mathcal{G}_{EA}$ | $\mathcal{G}_T$ | $\mathcal{G}_D$ | Succ ↑ | SPL ↑ | Succ ↑ | SPL ↑ | Succ ↑ | SPL ↑ |
| --- | --- | --- | --- | --- | --- | --- | --- | --- | --- |
| Straight | ✗ | ✗ | ✗ | 100.00 | 99.75 | 100.00 | 99.62 | 100.00 | 99.57 |
| | ✓ | ✗ | ✗ | 99.90 | 99.05 | 98.20 | 95.06 | 95.91 | 90.53 |
| | ✓ | ✓ | ✗ | 79.40 | 73.48 | 71.30 | 67.48 | 62.16 | 58.06 |
| | ✓ | ✓ | ✓ | 68.00 | 61.62 | 49.10 | 44.56 | 23.45 | 18.84 |
| Curved | ✗ | ✗ | ✗ | 100.00 | 97.62 | 100.00 | 97.47 | 100.00 | 98.18 |
| | ✓ | ✗ | ✗ | 99.70 | 95.93 | 97.50 | 90.14 | 89.30 | 79.95 |
| | ✓ | ✓ | ✗ | 65.00 | 56.70 | 58.10 | 52.51 | 47.70 | 42.28 |
| | ✓ | ✓ | ✓ | 35.50 | 18.38 | 23.90 | 12.08 | 12.50 | 6.84 |

## 5.2 Training on Passive Videos in the Wild

Finally, we demonstrate that our model can be learned from passive videos in the wild. Towards this end, we train our NRNS model using the RealEstate10K dataset [40] which contains YouTube videos of real estate tours. This dataset has 80K clips with poses estimated via SLAM. Note that the average trajectory length is smaller than test time trajectories in Gibson or MP3D, and we therefore only evaluate on 'easy' and 'medium' settings. Tab. 4 shows the performance. Note there is a drop in performance compared to training using Gibson videos, which we attribute to domain shift. Even after this drop, our approach, trained on real-world passive videos, outperforms BC baselines and performs competitively against RL baselines. RL baselines have an advantage as they are both trained in the simulator and trained on the Gibson dataset. We believe that the delta between RL and the performance of NRNS trained on RealEstate would close if both were tested on real-world data. We find these results to be a strong indication of the effectiveness of training on passive data.

Table 4: Comparison of our model (NRNS) trained with different sets of passive video data and tested on Image-Goal Navigation on **Gibson** [41]. We report average Success and Success weighted by inverse Path Length (SPL) @ $1m$. Results shown are tested without sensor & actuation noise.

| Path Type | Training Data | Model | Easy | | Medium | |
| | | | Succ ↑ | SPL ↑ | Succ ↑ | SPL ↑ |
| --- | --- | --- | --- | --- | --- | --- |
| Straight | RealEstate10k [40] | NRNS | 56.42 | 48.01 | 30.30 | 25.67 |
| | MP3D | NRNS | 59.80 | 52.35 | 37.00 | 31.89 |
| | Gibson | NRNS | **68.00** | **61.62** | **49.10** | **44.56** |
| | Gibson | BC w/ ResNet + GRU | 30.20 | 29.57 | 12.70 | 12.48 |
| Curved | RealEstate10k [40] | NRNS | 21.10 | 15.76 | 12.90 | 5.57 |
| | MP3D | NRNS | 28.26 | 13.59 | 11.00 | 5.10 |
| | Gibson | NRNS | **35.50** | **18.38** | **23.90** | **12.08** |
| | Gibson | BC w/ ResNet + GRU | 3.10 | 2.61 | 0.80 | 0.77 |

## 6 Conclusion

We have presented a simple yet effective approach for learning navigation policies from passive videos. While simulators have become fast, the diversity and scalability of environments still remains a challenge. Our presented approach, NRNS, neither requires access to ground-truth maps nor online policy interaction and hence forgoes the need for a simulator to learn policy functions. We demonstrate that NRNS can outperform RL and behavioral cloning policies by significant margins. We show that NRNS can be trained on passive videos in the wild and still outperform all baselines.

## Acknowledgement

The authors would like to thank Saurabh Gupta for the discussions. We would also like to thank the Gibson and RealEstate10K dataset authors for sharing their datasets for scientific research.

**Licenses for referenced datasets.**
Gibson: `http://svl.stanford.edu/gibson2/assets/GDS_agreement.pdf`
Matterport3D: `http://kaldir.vc.in.tum.de/matterport/MP_TOS.pdf`

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
