# No RL, No Simulation: Learning to Navigate without Navigating

**Meera Hahn** *
Georgia Institute of Technology

**Devendra Chaplot**
Facebook AI Research

**Shubham Tulsiani**
Facebook AI Research

**Mustafa Mukadam**
Facebook AI Research

**James M. Rehg**
Georgia Institute of Technology

**Abhinav Gupta**
Facebook AI Research

# Appendix

## 1 Implementation Details

Our NRNS model is implemented in PyTorch [1]. We use the each datasets given train/val/test splits. We tune hyperparameters based on val-unseen split performance and use the checkpoint with the highest val-unseen split accuracy to test the NRNS agent on Image-Goal navigation.

### 1.1 Distance Prediction Network Implementation

$\mathcal{G}_D$ is a graph neural network followed by fully connected layers. The fully connected layers take in the pairwise concatenation of the outputted GNN node features and goal image feature. The output of the fully connected layers is the predicted distance to the goal of each node. We implement a graph attention (GAT) network with edge feature attention using the PyTorch Geometric library [2]. The GNN is composed of two GAT layers trained with dropout of .6. For both GAT layers, the input to the layer is node dimension 512, edge dimension 16 and the output dimension of the nodes is 512. The node features output by the second GAT layer are pairwise concatenated with the ResNet18 feature of the goal image. The concatenated features are fed into a 2-layer MLP (512 to 256 to 1) with ReLU activation, and the output is fed through a sigmoid. The network is trained with mean squared error (MSE) loss against the true distance to goal. The network is trained with the Adam optimizer and the learning rate is .0001. Training the $\mathcal{G}_D$ model takes 20-25 epochs, requiring $\sim$6 hours on a single GPU.

**Unexplored nodes do not have features.** Note unexplored nodes do not have features so the features of these nodes are set to zero. Since unexplored nodes do not have children (no outward edges) their empty features are not propagated to any other nodes and only receive propagated features from explored nodes. The graph encoder simply creates the graph adjacency matrix, encodes the explored node as visual 512 features using a ResNet18 via the observed images and sets the edge features to be the transformation matrix between poses of neighboring nodes.

**Distance Score Implementation.** In $\mathcal{G}_D$, the distance label is implemented as a score between 0 to 1 which equals the inverse of the step-wise distance from each node to the goal image calculated by $1 - max(distance, 30)/30$. This clipped inverse distance score prioritizes small distances in the loss calculation. During inference time, the distance from the agent's current location $n_t$ to an unexplored node is a added as a 'travel cost' to the distance prediction. The predicted distance score $d_i$ is first converted by to a step-wise distance before the travel cost is added. The $\mathcal{G}_D$ network is trained with MSE loss over the predicted and ground truth distance scores. Additionally, loss is only back-propagated over the predictions on unexplored nodes.

---

*Correspondence: meerahahn@gatech.edu

## 1.2 Target Prediction Network Implementation

Input to $\mathcal{G}_T$ is the 512 dimension ResNet18 image features for the current and goal images. These features are fed through a single linear layer (512 to 512) followed by a ReLu activation. The hadamard product of the two vectors then fed into a linear layer (512 to 256) followed by a ReLu activation. The output is fed into 3 separate linear layer heads (256 to 1) (the output of these layers are the distance, rotation and switch predictions). The output of the switch linear layer is passed into a sigmoid function. The network is trained using a total loss of the sum of the Smooth l1 loss over the distance and rotation outputs and a Binary Cross Entropy (BCE) loss over the switch predictions. The network is initialized with Xavier uniform and trained with the Adam optimizer, learning rate=.0001 and dropout=.25 after the 2nd linear layer. Training the $\mathcal{G}_T$ model takes 10-15 epochs, requiring $\sim$2 hours on a single GPU.

## 1.3 RL Baseline Implementation

We use the Habitat [3] implementation of DDPPO [4] and follow the standard parameters. We train our DDPO agent on 8 GPUs. During training of the agent receives a terminal reward $r_T = 2.5$SPL, and shaped reward $r_t(a_t, s_t) = -\Delta_{\text{geo-dist}} - 0.01$, where $\Delta_{\text{geo-dist}}$ is the change in geodesic distance to the goal by performing action $a_t$ in state $s_t$. The RL baselines were evaluated over 3 random seeds and the average of the 3 runs was reported.

# 2 Videos in the Wild

## 2.1 RealEstate10k Dataset Description

RealEstate10K [5] is a large video dataset of trajectories through mostly indoor scenes. 80k video clips, containing $\sim$10 million frames each corresponding to a provided camera pose. The poses are procured from SLAM and bundle adjustment algorithms run on the videos, and they represent the orientation and path of the camera along the trajectory. The clips are gathered from 10k YouTube videos of real estate footage. The clips are relatively short and range between 1-10 seconds [5]. While the total number of frames in the RealEstate10k clips is large, the total length of the trajectory in meters is on average shorter than the MP3D and Gibson videos. Figure 1 shows the visual difference between frames of the passive video dataset created from the simulator and those taken from YouTube videos.

## 2.2 Passive Video Transfer Results on MP3D

On the MP3D dataset [6], we perform similar experiments as described in Section 5.2 of the main paper. This section of the main paper only showed results of the experiments on the Gibson dataset [7].

The results of these experiments again demonstrate that NRNS can be learned from passive videos in the wild. We use the same NRNS model trained on RealEstate10K [5] dataset as in Section 5.2 and test on the MP3D test split. Additionally we test an NRNS model trained on passive videos from the Gibson train split, and report performance on MP3D test split.

We find that training on passive videos from the simulator outperforms training on the passive videos on RealEstate10K. This can be attributed a few domain gap factors. RealEstate10K videos are significantly shorter than the simulator generated passive videos, resulting in less training data for the distance prediction network. Additionally, the simulator generated passive videos contain the same actions for the agent's rotation and translation as in the navigation task, where as RealEstate10K contains a different action space. Despite these domain transfer challenges the NRNS model trained on wild passive videos is able to outperform all other baselines.

# 3 NRNS Ablation Results on MP3D

We present results of the NRNS ablation experiments on MP3D [6]. The ablation experiments here are identical to those described in Section 5.1, of the main paper, for which performance is shown in Table 3, of the main paper, on Gibson [7]. We observe similar patterns in the NRNS ablations

Table 1: Comparison of our model (NRNS) trained with different sets of passive video data and tested on Image-Goal Navigation on **MP3D** [6]. We report average Success and Success weighted by inverse Path Length (SPL) @ $1m$. Results shown are tested without sensor & actuation noise.

| Path Type | Training Data | Model | Easy | | Medium | |
|---|---|---|---|---|---|---|
| | | | Succ ↑ | SPL ↑ | Succ ↑ | SPL ↑ |
| Straight | RealEstate10k [5] | NRNS | 44.58 | 39.27 | 15.81 | 10.73 |
| | Gibson | NRNS | 59.20 | 54.12 | 22.90 | 19.34 |
| | MP3D | NRNS | **64.70** | **58.23** | **39.70** | **32.74** |
| | MP3D | BC w/ ResNet + GRU | 30.20 | 29.57 | 12.70 | 12.48 |
| Curved | RealEstate10k [5] | NRNS | 9.43 | 4.96 | 5.30 | 2.86 |
| | Gibson | NRNS | 12.10 | 5.66 | 8.48 | 4.92 |
| | MP3D | NRNS | **23.70** | **12.68** | **16.20** | **8.34** |
| | MP3D | BC w/ ResNet + GRU | 3.10 | 2.61 | 0.80 | 0.77 |

results on MP3D as on Gibson. We again see that the Global Policy $\mathcal{G}_D$ has the greatest effect on performance out of all modules particularly on episodes with more difficult settings.

Table 2: Ablations of NRNS with baselines on Image-Goal Navigation on **MP3D** [6]. We report average Success and Success weighted by inverse Path Length (SPL) @ $1m$. ✗ denotes a module being replaced by the ground truth labels and a ✓ denotes the NRNS module being used.

| Path Type | NRNS Ablation | | | Easy | | Medium | | Hard | |
|---|---|---|---|---|---|---|---|---|---|
| | $\mathcal{G}_{EA}$ | $\mathcal{G}_T$ | $\mathcal{G}_D$ | Succ ↑ | SPL ↑ | Succ ↑ | SPL ↑ | Succ ↑ | SPL ↑ |
| Straight | ✗ | ✗ | ✗ | 100.00 | 100.00 | 99.80 | 99.07 | 100.00 | 99.02 |
| | ✓ | ✗ | ✗ | 100.00 | 100.00 | 99.00 | 96.81 | 96.10 | 92.65 |
| | ✓ | ✓ | ✗ | 74.20 | 68.19 | 64.50 | 60.13 | 58.40 | 55.38 |
| | ✓ | ✓ | ✓ | 64.70 | 58.23 | 39.70 | 32.74 | 22.30 | 17.33 |
| Curved | ✗ | ✗ | ✗ | 100.00 | 94.08 | 99.90 | 95.39 | 100.00 | 97.00 |
| | ✓ | ✗ | ✗ | 100.00 | 93.06 | 97.70 | 90.22 | 91.60 | 82.87 |
| | ✓ | ✓ | ✗ | 62.20 | 53.31 | 54.10 | 47.51 | 51.00 | 44.92 |
| | ✓ | ✓ | ✓ | 23.70 | 12.68 | 16.20 | 8.34 | 9.10 | 5.14 |

## MP3D Passive Video Examples

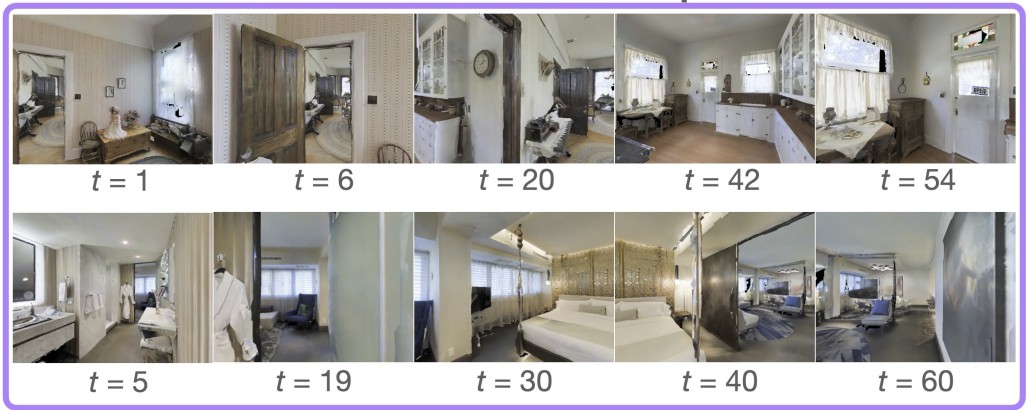

## Gibson Passive Video Examples

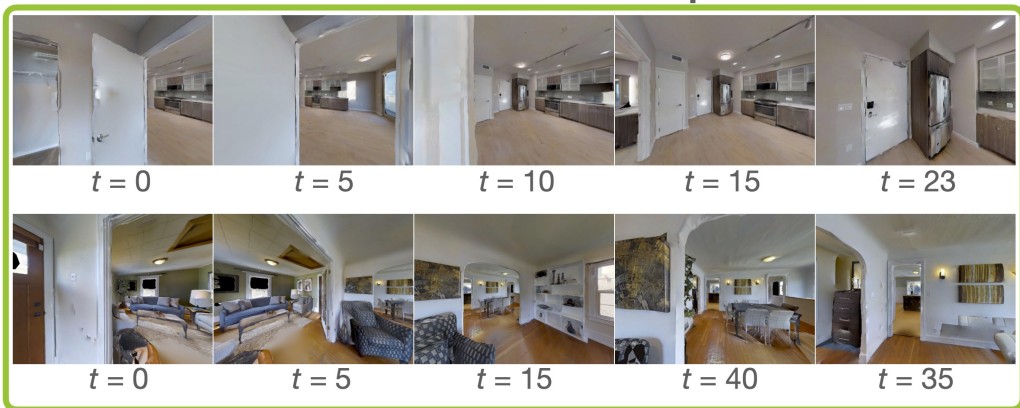

## RealEstate10k Passive Video

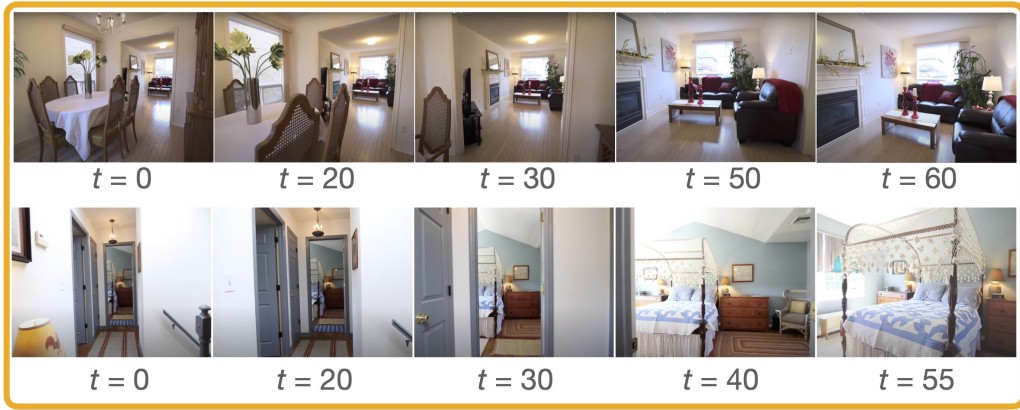

Figure 1: Comparison of the passive videos from different datasets used for training our NRNS agent. MP3D and Gibson passive video frames are images of rendered environments using the habitat simulator and therefore are similar in photo realism. RealEstate10K video frames are taken directly from a real estate tour YouTube video and therefore differ from MP3D and Gibson.