# OpenReview forum: "No RL, No Simulation: Learning to Navigate without Navigating"
_NeurIPS.cc/2021/Conference — NeurIPS 2021 Poster_

### Official Review · Reviewer_szNL · 2021-07-12

**Rating:** 7
**Confidence:** 4

**Summary:**

This work details a method for planning and navigation on the image-driven navigiation task in a photo-realistic simulator. The novelty of this approach is that the method is trained with sequences of (RGBD, pose) sampled from the environment, without the associated actions or rewards. The sequences are used to learn a predictive model of a distance function, which the authors claim can learn priors about the indoor layouts of indoor buildings, which enables the method to be transferred to buildings that were not observed during training.


**Limitations And Societal Impact:**

One limitation is the requirement of pose estimates. The authors show a comparison where nose is added to the pose estimates. A more real world comparison where SLAM is used directly on the sequences to infer relative position and orientation would be interesting, but is perhaps beyond the scope of an already extensive conference paper.

A second interestesting extension would be sim2real transfer of policies trained with such a method.

**Main Review:**

The work presented is novel, of high quality and clearly written. The method is particularly of interest as it can be used to create a policy without the need to interact in a simulator, which means the method has the potential to scale to vast real world datasets.
The method is compared against three baselines and outperforms them by a large margin. Detailed ablations are shown to break down the modular approach of this method.
Whilst the approach is novel, and outperforms the presented baselines by a large margin, I have a number of questions about the details of the implementation of the baselines.

Strengths:
1. The work outperforms baselines based on RL and behavioral cloning.
2. This approach decouples low-level acting and planning in interpretable manner .
3. The modular approach means that each component can be trained independently, which separates the problem into independant subproblems.

Weakness:

Baselines - I have a number of questions and concerns about the baselines:
0. It would have been good include a baseline with a policy of sampling actions randomly, in order to ensure the other baselines outperform purely random actions.
1. The baseline performs rather poorly in this setting, whilst the task is challenging I still find a success rate of 3.6% on goals within 1.5-3m suprisingly low.
2. How many seeds were evaluated when training the baseline method?
3. What is the reward function used for the RL method?
4. Input resolution: Given the instability and optimization challenge of training an RL method, an input resolution of 480x640 seems rather large, have the authors tested a down-sampled input?
5. On-policy Deep-RL methods such as PPO are known to be sample inefficient, do the authors consider the baseline to have converged after a "mere" 10M environment interactions?
6. Stopping criterion, it would be interesting to see comparisons of the baseline methods using the learned stopping criterion. As one reason an end to end RL method performs poorly in the setting could be related to the difficulty of exploring the action space when a STOP action is present, which presumable terminates the episode. In Neural Topological SLAM the approachs were compared with and without the need to a STOP actions. It would be good to have included this comparison.
7. Many recent works have shown scaling and distributed training of RL methods can outperform handcrafted learning based methods, for example [1] vs [2]. What is the authors opinion of scaling RL methods on a more complex task such as image-driven goal navigation?

Handcrafted learning based methods require time, tuning and domain expertise. How transferable is this approach to other tasks such as object goal navigation or coverage maximization?

For the local navigation policy, have the authors compared against an local pointgoal RL agent?

Missing references:
I draw the authors attention to [3] which is a recent related work on topological planning with graph neural networks in photorealistic envrionments.

[1] DD-PPO: Learning Near-Perfect PointGoal Navigators from 2.5 Billion Frames. Wijmans et al. ICLR 2020
[2] LEARNING TO EXPLORE USING ACTIVE NEURAL SLAM. Chaplot et al. ICLR 2020
[3] Learning to plan with uncertain topological maps. Beeching et al. ECCV 2020

UPDATE AFTER AUTHOR RESPONSE: Thank you to the authors for addressing my questions. I will leave my review at 7.

**Time Spent Reviewing:**

2

---

> ### Author Response · Authors · 2021-08-10
> **Response to szNL**
>
> 1.  *Providing a random baseline*: We provide a random baseline below -- ran with 3 random seeds and averaged. In this baseline the agent randomly selects out of the actions -- turn left, turn right, go forward, STOP -- at each step. The Succ and SPL are 0.0 for both Medium and Hard settings for both datasets. The results on the Easy setting:
>
> Random Baseline - Mp3d
>
> | Episode Type  | Success | SPL |
> | ------------- | ------------- |  ------------- |
> | Easy-Straight  | 0.0260 |  0.0258 |
> | Easy-Curved  | 0 0.0030 | 0.0030 |
>
> Random Baseline -Gibson
>
> | Episode Type  | Success | SPL |
> | ------------- | ------------- |  ------------- |
> | Easy-Straight  | 0.0180 |  0.0177 |
> | Easy-Curved  | 0.0040 | 0.0040 |
>
>
> 2. *RL baselines: implementation details, convergence on 10m steps and thoughts on scaling*: We do not believe the RL algorithm has converged at 10M steps and would improve with more training. However in the context of scalability, approaches like NRNS can also scale in performance in respect to diversity and size of training data. So, our current evaluation is at an operating point which is similar on both axes for all methods, similar compute and diversity to both approaches.
>    However, to highlight the fact that even with significantly more compute RL-based approaches struggle. We have improved the RL-baseline by increasing the diversity of data (sampling more initial and goal states aka increasing the number of training episodes per environment), increasing batch size and adding more layers to LSTM. Specifically, we train a Habitat implementation of DDPPO and we use 8 gpus and 12 processes per gpu allowing for a larger batch size. We also increased the number of LSTM layers from 2 to 3. Finally, we provide RL 5x more data (number of training episodes per environment) than NRNS. We report the performance at 50M (5x more compute than NRNS) and 100M steps (10x more compute).
>
>     At 50M, RL shows an average success rate of 25.98% for straight and 12.33% for Curved. At 100M, RL shows an average success rate of 29.01% for Straight and 15.7% for Curved. Even though NRNS is at both data (lower number of training episodes per environment) and compute disadvantage, it has performance of 47% success for Straight and 24% success for Curved test trajectories. As the results indicate, the performance improves but is still lower than NRNS. We believe if we train more, RL approaches are likely to further improve but with diminishing returns.
>
> 3. *How many seeds were evaluated when training the baseline method?* All the baselines were evaluated over 3 random seeds and the average of the 3 runs was reported.
>
> 4. *What is the reward function for the RL method?* Follow prior work the agent receives terminal reward rT = 2.5 SPL, and shaped reward $rt(at, st) = −\Delta geodist − 0.01$, where $\Delta geodist$ is the change in geodesic distance to the goal by performing action at in state st
>
> 5. *Experiment with Oracle stop*: We simulated this experiment on NRNS. Table 4 shows ablation with ground truth for the $\mathcal{G}_{T}$ function. We can add the oracle stop ablation for the Behavior Cloning and RL baselines as well to the final paper.
>
> 6. *Is NRNS transferable to other tasks*: We believe so! We believe NRNS should provide a strong baseline for other navigation tasks and even manipulation tasks. For example, in object goal navigation, we would be reliant on object annotations in a video or reliant object recognition system. Using human or shelf annotations one could create subgraphs and train NRNS hierarchical approach. But this is beyond scope of paper and a direction for future research.
>
> 7. *For the local navigation policy, have the authors compared against a local pointgoal RL agent?* We haven’t compared with a local point-goal RL agent as the local policy is not our contribution. It is adapted from NTS [4] and ANS [10].
>
> 8. *Missing references: I draw the authors attention to [3]*: Thank you for bringing this paper to our attention, we will include reference in our paper.

---

### Official Review · Reviewer_dPyV · 2021-07-17

**Rating:** 8
**Confidence:** 4

**Summary:**

The paper introduces a new method for learning a navigation policy for indoor environments without RL and without simulation (which is, admittedly, fitting given the title). The method uses SLAM on passive video recordings to construct navigation graphs that are then used to train several networks including a goal-detection network that can be used in goal-based navigation.

I personally like the idea in this paper a lot and clearly, a lot of work went into the experiments and into the polish. That is why I'm upset that I have to give this paper a lower score than it deserves. The main thing that's hindering me from wholeheartedly recommending this is the lack of code and reproducibility. If code is not included, then I have to put the methods section under more scrutiny and there aren't enough details to reproduce this work. I'm happy to change my score if the authors address the issues listed below.

**Limitations And Societal Impact:**

The authors have neither assessed limitations nor possible societal impact in the main body of the paper.

Here are some ideas to discuss for limitations:
- The work is reliant on the video preprocessing. Right now this was demonstrated with video that's in-distribution and slightly out-of-distribution but not with video "in the wild". Given how sharply the performance drops when using non-Gibson data to train a Gibson agent, how would you expect this to perform with random youtube videos?
- Do you consider the task solved with a 68% success rate in straight 1.5-3m navigation tasks?

For societal impact: This work, like many others, enables indoor navigation agents. That could lead to the development of more capable robotic assistants in the long run, especially since this work aims to lower the bar for entry by removing simulation and RL.

**Main Review:**

**Originality:**

There are many works on using SLAM as part of an RL pipeline but all of these need to be trained interactively. Other works do indoor navigation from imitation and these, too, often need to be tuned via RL. The works that do learn from passive video, don't learn a distance estimator and only compare the current frame of a trajectory to the goal. Therefore I think the method presented in this work is sufficiently novel.


**Quality:**

The experiments sufficiently back up the method, the method is well-motivated, the ablations are insightful.
The only concerns I have with respect to quality are
1. The low performance of PPO - without listing the hyperparameters. I have used PPO for indoor navigation in Gibson for my own experiments and with a bit of tweaking, the performance can be brought up to a 10-15% success rate, especially on the "easy" task. I would recommend the authors spend a little more time with this baseline to establish a fair comparison.
2. Why is the agent with access to the metric map so bad? I don't think this was sufficiently justified in the paper.
3. In the supplemental video, why does the second trajectory just fade out long before reaching the target? And the third trajectory seems to miss the target. I'm very confused by the video - it makes your method look significantly worse.


**Clarity:**

As mentioned above, this is my main concern with this paper. Again, I'm only this strict because you didn't include any code with your submission. I can't accept any promise to submit the code later since I can't review non-existent future code.
There are 4 subcomponents to your method,
- $G_D$, the global policy
- $G_{LP}$, the local policy
- $G_{T}$, the stop signal function
- $G_{EA}$, the explorable area function

$G_D$ is learned with a Graph Neural Network. There is a nice plot of the network in Figure 3 but is that all? What are the sizes of the GAT layers, their inputs and outputs? Is the MLP single-layer or multi-layer. You wrote "simple MLP" but what does that mean? Could we maybe get a network table where each layer is listed, together with their activation function and the number of parameters? And what exactly is the structure of the graph encoder?
And for preparing the data, you wrote "First, we generate a stepwise trajectory based on odometry information from each frame." (line 210)... How exactly do you do that? What kind of SLAM system is employed here, and with which parameters? And likewise, "We adapt this trajectory to topological graph structure using affinity clustering on the stepwise graph via visual features and odometry information." - How? Which parameters, and which affinity clustering algorithm is used? Without this information, I cannot reproduce this.

$G_{LP}$ is "adapted from [10]" - but what does that mean? "Adapted" means "make (something) suitable for a new use or purpose; modify", so how was it modified to suit your needs?

"$G_{T}$ is simple MLP" (sic! line 186) - what does that mean? How many layers, activation functions, etc? Apparently, it has multiple output heads (for goal in sight and relative position of goal).

$G_{EA}$ - "This is untrained, heuristic function" (sic! line 175). It supposedly returns angles that don't have an obstacle within 3m. But what if there is a partial obstacle? Say half the frame has an obstacle, or a third or a fifth? What's the cutoff? Or what if the obstacle is 2m above ground like a doorframe?

Also, from the text, line 148, you are using the visual features of unexplored nodes in your $G_D$. How does that work - which network generates visual features of unexplored nodes?

And in general, what are any of the loss functions you are training, what is your optimizer, what are the training parameters? There are superficial details in chapter 1 of the appendix, but the losses aren't included there and the detail there is not enough to warrant any reproduction.



**Significance:**

I think if it were reproducible, this would be a significant contribution.

**Nitpicks and Questions:**

1. The first third of the paper is well-written and mostly error-free. But as pointed out above, sections 4 and 5 are riddled with errors like missing articles and missing commas. Please proofread the document! For example lines 180, 228, 239, 259 twice, 279, 290, 305. Also, I believe that in general, if you write "x-based" like RL-based or Graph Neural Network-based, there needs to be a dash between "based" and the word before.
2. What's the PPO implementation and backend? I assume you're not using the standard Atari CNN, right?
3. In Table 1, for a 1.5-3m goal that's in a straight line from the agent, why is the performance so bad?

**Time Spent Reviewing:**

8

---

> ### Author Response · Authors · 2021-08-10
> **Response to dPyV**
>
> *Reproducibility*: We acknowledge R3’s need for reproducibility. We are fully committed to releasing our code with a camera ready version.  As a show of good faith and to provide a source for implementation details we have provided an anonymous link (https://github.com/neuripsSubmission/Paper4346) to our code. The code in its current state is not yet user friendly but as part of our future code release we will be cleaning up this code base and uploading all the models and data files. That said, we answer all questions on implementation and will add these details to the paper.
>
> **_Quality Questions_**:
> 1. *The low performance of PPO, parameters and implementation*: We use the default habitat implementation of PPO, which contains a ResNet18 and LSTM with 2 layers. The habitat implementation of PPO and the image nav config files can be found here: https://github.com/facebookresearch/habitat-lab/tree/master/habitat_baselines. We will add all these details to the paper.
>
>     We do not believe the RL algorithm has converged at 10M steps and would improve with more training. However in the context of scalability, approaches like NRNS can also scale in performance in respect to diversity and size of training data. So, our current evaluation is at an operating point which is similar on both axes for all methods, similar compute and diversity to both approaches.
>
>      However, to highlight the fact that even with significantly more compute RL-based approaches struggle. We have improved the RL-baseline by increasing the diversity of data (sampling more initial and goal states aka increasing the number of training episodes per environment), increasing batch size and adding more layers to LSTM. Specifically, we train a Habitat implementation of DDPPO and we use 8 gpus and 12 processes per gpu allowing for a larger batch size. We also increased the number of LSTM layers from 2 to 3. Finally, we provide RL 5x more data (number of training episodes per environment) than NRNS. We report the performance at 50M (5x more compute than NRNS) and 100M steps (10x more compute).
>
>    At 50M, RL shows an average success rate of 25.98% for straight and 12.33% for Curved. At 100M, RL shows an average success rate of 29.01% for Straight and 15.7% for Curved. Even though NRNS is at both data (lower number of training episodes per environment) and compute disadvantage, it has performance of 47% success for Straight and 24% success for Curved test trajectories. As the results indicate, the performance improves but is still lower than NRNS. We believe if we train more, RL approaches are likely to further improve but with diminishing returns.
>
> 2. *Performance of BC w/ metric map*:
> We find the ResNet18 + Metric Map + BC performs worse than ResNet18 + GRU + BC. We attribute this to a spatial memory (metric map) being less informative for agent exploration than a memory of the visual features and previous steps (GRU).
>
> 3. *In the supplemental video, why does the second trajectory just fade out before reaching the target? Third trajectory seems to miss the target.*
> We acknowledge the supplemental videos require additional information. The Mp3d scans only show the graph nodes and not each individual step of the agent. In the third video the agent passes the location target by .5 meters but does not miss the target image as you can compare the final image of the agent and the goal image. To clarify this confusion have generated new videos which are downloadable at this anonymous link https://github.com/neuripsSubmission/Paper4346/tree/master/demo_videos which now show both the graph AND the intermediate steps of the agent between the nodes. Coloring of the nodes in the supplemental and linked videos is: Green = Start; Red = Goal, Yellow = Explored, White = Unexplored, Blue = Current position of agent.
>
> **_Clarity Questions_**:
> 1.  *What is the implementation of the network of $\mathcal{G}_{D}$? [L 148], which network generates visual features of unexplored nodes?*
> $\mathcal{G}_{D}$  is a graph neural network followed by fully connected layers. The fully connected layers take in the pairwise concatenation of the outputed GNN node features and goal image feature. The output of the fully connected layers is the predicted distance to the goal of each node. We implement the GAT with edge feature attention using the TorchGeometric library. The GNN is composed of two GAT layers trained with a dropout of .6. For both GAT layers, the input to the layer is node dimension 512, edge dimension 16 and the output dimension of the nodes is 512. The node features output by the second layer GAT are pairwise concatenated with the ResNet18 512 goal image feature. These new features are fed into a 2 layer MLP (512 to 256 to 1) with ReLU activation, the output is fed into a sigmoid. The output of the network is trained with MSE loss against the true distance to goal. Optimizer is Adam and the learning rate is .0001. Please see https://github.com/neuripsSubmission/Paper4346/blob/bd2b3b419884d63f0cca6869800b8b497b13321c/src/functions/distance_func/deepgcn.py#L12 for the model definition.
> Unexplored nodes do not have features so the features of these nodes are set to zero. Since unexplored nodes do not have children (no outward edges) their empty features are not propagated to any other nodes and only receive propagated features from explored nodes. The graph encoder simply creates the graph adjacency matrix, encodes the explored node as visual 512 features using a ResNet18 via the observed images and sets the edge features to be the transformation matrix between poses of neighboring nodes. All of these details will be added to the paper.
> 2. *"First, we generate a stepwise trajectory based on odometry information from each frame." [L 210] How exactly do you do that?*
> For the Gibson and MP3D dataset passive trajectories we take the ground truth odometry information that the simulator produces. For the RealEstate10k dataset we use the dataset provided odometry information which is produced via SLAM and bundle adjustment see [32] for details. We then employ the Affinity Propagation Clustering over the node features which are concatenated ResNet18 image features and node pose. We use the sklearn implementation of Affinity Propagation Clustering with random_state=5 and the default parameters.
> 3. *GLP  is "adapted from [10]" - how was it modified to suit your needs?*
> Two changes were made from [10]. The first is that our input depth image was 120 degree FOV not panoramas with 360 degree FOV, which resulted in a more limited metric map. Secondly [10] does not take into account the orientation of the agent upon reaching the sub-goal location. We modify the method to also turn the agent to match the sub-goal orientation upon reaching the sub-goal location.
> 4. *Describe the implementation details of $\mathcal{G}_{T}$:*
> Input to $\mathcal{G}_{T}$ is 512 dimension ResNet18 image feature for the current and goal images, These images go into a single linear layer (512 to 512) followed by a ReLu, The two vectors are multiplied and the output is fed into a linear layer (512 to 256) followed by a ReLu. The output is fed into 3 separate linear layers (256 to 1)  (the output of these layers are the distance, rotation and switch predictions). Output of the switch linear layer is passed into a sigmoid function. The network loss function is the sum of smooth l1 loss over the distance and rotation outputs and a bce loss over the switch predictions. Initialized with Xavier uniform and trained with the Adam optimizer, learning rate=.0001 and dropout=.25 after the 2nd linear layer. Please see https://github.com/neuripsSubmission/Paper4346/blob/bd2b3b419884d63f0cca6869800b8b497b13321c/src/functions/target_func/goal_mlp.py#L10 for the model definition.
> 5. *How does $\mathcal{G}_{EA}$ handle partial obstacles or high up obstacles like a doorframe?*
> Partial obstacles will affect the agent's ability to traverse forward at different angles. GEA only looks at obstacles up to the height of the agent (1.5m).
> 6. *Loss and general implementation*:
> Please see answers to above Q1, Q3. The loss for $\mathcal{G}_{T}$ a sum of smooth l1 loss for the distance and rotation output and bce loss over the switch output. $\mathcal{G}_{D}$ is trained with an mse loss. Both functions are trained with adam optimizer.
>
> **_Nitpicks and Questions_**:
> 1.  *Writing errors, grammar and typos*:
> We thank the reviewer for pointing out the writing issues and we will take a thorough proofread of the document.
> 2. *Performance on ‘straight-easy’ episodes?*
> Straight line refers to shortest paths with a difference in heading between the start and end positions to be < 45°. Meaning the agent does not have to turn around significantly. However this does not mean that the goal in-sight from the beginning of the episode and therefore it is episodes can be nontrivial. Additionally agents can sometimes over/undershoot the target due to the number of matching features in their FOV to the goal image.
>
> **_Limitations And Societal Impact Questions_**:
> 1.  *How would you expect NRNS to perform with random youtube videos?*
> Table 4 shows when trained on RealEstate10k which is a youtube video dataset of random houses and we do see a drop in performance. The data is clearly very important in terms of length of the videos and the number of videos overall and types of house in the videos -- training on only office buildings and testing on family houses will probably have a drop in performance and due to semantic differences in the environment. But the key is that obtaining videos is much more scalable than building simulators.
> But what makes our approach appealing is the fact: when it comes to testing in the real world and on a real robot, our approach has a significant advantage since there is no domain gap and no need for sim-to-real.
> 2. *Is 68% success rate in straight 1.5-3m episodes solved?*
> No, we do not consider the problem to be solved.

---

> > ### Author Response · Authors · 2021-08-18
> > **Paper Rating Update**
> >
> > Dear Reviewer dPyV
> >
> > You mentioned that you “like the idea in this paper” and “ upset that I have to give this paper a lower score than it deserves” due to code and reproducibility.
> >
> > In our rebuttal, we have anonymously opensourced the code and provided responses to all your reproducibility queries. We were wondering if you would be now comfortable in reevaluating the paper in this light and provide new rating.
> >
> >  Thanks
> > Authors

---

> > ### Comment · Reviewer_dPyV · 2021-08-19
> > **Author response above and beyond, well done! Increasing score to 8.**
> >
> > Dear Authors,
> >
> > I just have to say, your response and the additional work you put in are excellent!
> > I particularly appreciate the code (obviously it needs documentation but I've seen worse research code) and the new demo videos. Also, the additional PPO results are appreciated.
> > Since you addressed all my concerns, I can now wholeheartedly recommend this work and will change my rating accordingly.
> > I hope that you incorporate the implementation details that you wrote above in your `Clarity Questions` response into the paper.
> >
> > Best,
> > Reviewer dPyV

---

### Official Review · Reviewer_ncbG · 2021-07-17

**Rating:** 7
**Confidence:** 5

**Summary:**

The paper addresses image-goal navigation, where an agent is placed in a novel environment, and is tasked with navigating to an (unknown) goal position that is specified using an image taken from that position. The method builds a topological map and uses three learned networks for planning navigation. The first one, G_D, is a GNN which takes the current topological graph and goal image, and predicts distances to goal for all nodes. The second one predicts visual features for unexplored nodes in the graph. The third one, G_T, is a termination predictor which estimates probability of goal within sight, and its relative pose. Finally, the system uses non-learned components: shortest distance planner, a local heuristic navigation policy G_LP for reaching subgoals, a heuristic module G_EA for expanding graph with unexplored nodes. The experiments are conducted in 2 environments: Gibson and MP3D. The method shows improvement with respect to 2 BC and 1 RL baseline.

**Ethical Concerns:**

No ethical issues with this paper.

**Limitations And Societal Impact:**

The authors adequately addressed the limitations and potential negative societal impact of their work.

**Main Review:**

The scores below are consistent with "Rating" legend.

## Originality = 5/10
The proposed method is new. However, the paper would profit from a better embedding into the prior work. From the title "No RL, No Simulation: Learning to Navigate without Navigating" a reader could make a conclusion that it is the first time a passively learned navigation policy has shown success. However, there is prior work which already attempted to do this, albeit in a slightly more restricted setting. For example, SPTM [18] did learn its R and L networks passively, from data sampled by a random policy - so no RL-like feedback loop. It could also do it without a simulator, if it chose to have a local heuristic control policy instead of L-network. Of course, SPTM differs in test-time assumptions: it requires a short video of the test environment to begin with. This is the scientific "delta" of the proposed algorithm: it gets rid of this requirement by using a smart heuristic module G_EA for expanding the graph, and by a network which predicts visual features of newly expanded nodes. This is a nice research progress, but it requires a proper explanation of what is really new in this paper. If the authors choose to make such updates, I'm ready to increase my score.

## Quality = 7/10
The quality of experiments in this paper is high. There are enough experiments to support the main claims and there are exhaustive ablations to determine the value of each introduced component. Perhaps, one nice-to-have additional experiment would be to add a random walk baseline, and also SPTM + random-walk exploration.

## Clarity = 7/10
The paper is written in a nice and clear manner. A few questions though:
1. What are step limits for RL and BC?
2. [L111-112] "egocentric pose estimate" of what with respect to which origin?
3. [L166-167] "The local metric maps are discarded upon reaching the sub-goal" - how to determine that the subgoal is reached? What if it's not reached in reality? What if we think we reached on subgoal, but instead we reached another visually similar subgoal (i.e. perceptual aliasing)?
4. Could you please use bold font in tables to highlight the best number?

## Significance = 6/10
The submission does advance state-of-the-art by getting rid of annoying assumptions in prior work. The slight concern though is that the real frontier now lies more in real-world applicability of these algorithms than in conceptual improvements in simulated environments, because there have already been many navigation papers in simulation. "Scaling Local Control to Large-Scale Topological Navigation" https://arxiv.org/pdf/1909.12329.pdf scaled more restricted SPTM-like ideas to the real world. I would be curious to see if the ideas in this submission scale to the real world as well.

**Time Spent Reviewing:**

3h

---

> ### Author Response · Authors · 2021-08-10
> **Response to ncbG**
>
> **_Originality Rebuttal_**:
>
> 1. *Connections to SPTM*: We agree that the difference in SPTM setup and our setup is that SPTM requires a short video of the test environment while in our setup no experience in the test environment is required. However, we respectfully disagree that the setups are “delta” different. In our opinion, this assumption completely changes the problem, in SPTM the challenge is to localize the agent and target image in the video trajectory and follow directions in the video (note that SPTM is unable to tackle the case where the goal image is not seen in the video trajectory). In our setup, the challenge is to understand what the goal image depicts (what room? What objects?), use learned semantic priors on where the room or the objects are more likely to be found and efficiently explore the completely unseen environment to find the target image.
>
>
>     The parallel to our work is in the use of topological maps (which we agree with [18] that topological graphs are best suited for the navigation task) and in the construction of graphs from video footage. However, in our approach, the construction of graphs from video footage is not ever used in testing, it is only used to create training data. Additionally, the graph creation is not the main contribution of our paper. Our contribution is rather the totality of the modular NRNS approach image goal navigation and the training NRNS via the footage instead of the ground truth mesh.
> That said, We agree with the reviewer that we should better contextualize our approach with respect to SPTM. We will include a detailed discussion on describing how SPTM uses test videos to generate graphs and how it is different from our setup.
>
> **_Quality Rebuttal_**:
>
> 1.  Providing a random baseline.
>
> We provide a random baseline below -- ran with 3 random seeds and averaged. In this baseline the agent randomly selects out of the actions -- turn left, turn right, go forward, STOP -- at each step. The Succ and SPL are 0.0 for both Medium and Hard settings for both datasets. The results on the Easy setting:
>
> Random Baseline - Mp3d
>
> | Episode Type  | Success | SPL |
> | ------------- | ------------- |  ------------- |
> | Easy-Straight  | 0.0260 |  0.0258 |
> | Easy-Curved  | 0 0.0030 | 0.0030 |
>
> ***
> Random Baseline -Gibson
>
> | Episode Type  | Success | SPL |
> | ------------- | ------------- |  ------------- |
> | Easy-Straight  | 0.0180 |  0.0177 |
> | Easy-Curved  | 0.0040 | 0.0040 |
>
>
> **_Significance Rebuttal_**:
>
> 1. *Real-World Transfer*: We agree with the reviewer that the next frontier is real-world navigation systems. We believe that our approach does make strides towards this. Although our evaluation is in a simulated setup to allow for comparisons to prior SOTA approaches (and due to practical considerations in COVID), our experiments (see Table 4) clearly show that the learning can occur using only real-world passive videos. In fact, we would argue that this ability to learn from passive in-the-wild data is key for the real-world applicability in diverse environments. Our approach would not require sim-to-real transfer since it is already trained in the real world.
>
> **_Clairity Questions_**:
>
> 1. *What are step limits for RL and BC?*: Baselines and NRNS all have the same following params: Maximum steps = 500, Forward step distance=.25, turn angle=15$^{\circ}$, FOV= 120$^{\circ}$
>
> 2. *[L111-112] "egocentric pose estimate" of what with respect to which origin?*: Egocentric pose estimate of the agent with the origin being the position at t=0. For the YouTube videos the poses were estimated using SLAM please see [32] for details.
>
> 3. *[L166-167] "The local metric maps are discarded upon reaching the sub-goal" - how to determine that the subgoal is reached? What if it's not reached in reality? What if we think we reached one subgoal, but instead we reached another visually similar subgoal (i.e. perceptual aliasing)?*:  This is correct; the agent might not reach the exact location of the selected subgoal, for this reason the sub-goal node in the graph and corresponding edges are updated with the pose sensors when local navigation terminates. The local navigation is based on point navigation within a few meters so issues with not reaching the subgoal would be more likely to do with obstacles, holes in the mesh, etc rather than perceptual aliasing.
>
> 4. *Could you please use bold font in tables to highlight the best number?*: Yes we will make this change.

---

> > ### Comment · Reviewer_ncbG · 2021-08-10
> > **Increasing score to 7**
> >
> > After the clarifications provided by the authors, I'm increasing the overall score to 7. Please update the camera-ready based on our discussion.

---

### Official Review · Reviewer_D3yc · 2021-07-22

**Rating:** 5
**Confidence:** 4

**Summary:**

The paper looks into offline data as the sole source of training actor policies for navigational agents. The proposed approach achieves this by a mix of learnable modules/networks and rule-based, heuristic ones. Moreover, the overarching goal is to reduce (to zero) the number of interactions in the simulator during training. The approach is compared with behavior cloning and end-to-end RL flat policy. The task chosen by the authors is image-goal navigation i.e. navigating to a specified image on data that they generate on the Habitat simulator.

**Limitations And Societal Impact:**

Yes

**Main Review:**

**[Originality] (+) Model-based approaches are still under-explored for Habitat-like simulators:**
The rendering of RGB images is a computation bottleneck for current embodied AI research. While previous works have looked at distributed training (Wijmans et al. 2020), accelerating simulation (Savva et al. 2019), auxiliary losses (Ye et al. 2020), etc., this work investigates the promise of model-based approaches and with heuristic components.

**[Clarity] (+) The approach is clearly explained and figures are informative:**
The explanation of the approach is done in a clear top-down way which makes understanding quite easy. Relevant background from existing work is included, making the manuscript self-sufficient. Moreover, the figures and associated captions illustrate the approach and idea very effectively.

----

**[Originality] (-) The paper doesn’t connect/compare to research on ‘Offline Reinforcement Learning’ and ‘Learning from Demonstrations’ (or Offline IL):**

Offline RL, as thoroughly studied by multiple works, is aimed at utilizing passively collected offline data and needs no additional online interaction with the environment/simulator. Kindly refer to (Levine et al. 2020 arXiv:2005.01643) for a detailed tutorial, discussion of baselines, and open problems. While the submission is applying offline learning to visual navigation tasks with some existing neat modules on topological mapping and expansion, it fails to connect to the most relevant literature and claims offline learning in interactive environments as their key novelty. As mentioned in L245 “our key contribution is the ability to learn navigation agents from passive data.” This has been studied in existing work. Qualifying the claims of the work could be considered.

Particularly relevant are (a) Kahn et al. 2020 (BADGR) that runs on self-supervised off-policy data gathered in real-world environments needing ‘no simulation’ or human supervision; (b) Fu et al. 2020 (D4RL) that includes offline CARLA visual simulator for autonomous driving. Please see Levine et al. 2020 for more references on navigation and robot manipulation. The reward in a dataset of offline demonstrations could be any metric inversely proportional to the distance between states (or nodes, in the case of this submission). Also, the distance estimator $\mathcal{G}_{D}$ appears very similar to a critic estimating the value of a given state.

Additionally, the research in ‘learning from demonstrations/observations’ that learns from a mix of pre-collected demonstrations and interactions is relevant to the work. Hester et al. 2018 (DQfD),  Vecerik et al. 2017 (DDPGfD), and Nair et al. ICRA 2018 are relevant works for comparisons and grounding.

Considering domain shifts and using web/internet data as studied by Chang et al. [23] and also touched upon in Sec. 5.2 is a clear novel direction.

Q. The reviewer requests the authors to clearly relate and contrast their _key contribution_ to the offline RL/IL literature pointed above. Also, kindly highlight novelty differences from [23] (that tackle real-world youtube videos instead of traversals of simuators) beyond using memory in distance prediction (L78)

**[Originality] Request the authors to clearly distinguish what components in the _approach_ are novel.**

Discussing contributions beyond learning from passive offline data, several components in the approach are overlapping existing work on navigation in visual environments. The reviewer requests the authors clearly list the key contributions of the _approach_ with respect to [4,10]?

**[Quality] (-) Concern about benchmark chosen for evaluation:**
Since the demonstrations or training data used in the work is generated within the Habitat simulator, it’s straightforward to collect the same for the well-studied objectgoal or pointgoal navigation datasets. These tasks have been hosted within AIHabitat + (Gibson/MP3D scenes) with challenges run for 2 years at CVPR 2020 and 2021 workshops. Evaluating a new idea (of pure offline learning) on well-studied benchmarks and in the presence of reported results of previous baselines would add more credibility and also encourages the associated community to adopt this new idea.

The reviewer shares the concern of using a (presently nonstandard) image-goal navigation task when it has no particular advantage for this study over the well-studied embodied tasks. While image-goal navigation task has been studied before by [4] and [15], both these worked introduce their own implementation of the task and their own data. This paper, again, introduces a new realization of the task and a new dataset. Hence, all the baselines reported in Tables 1 and 2 are implemented by the authors themselves. While introducing a dataset is a great feat, it would benefit from being grounded in performance to other datasets, accompanied by rigorous analysis, and tests for any data biases. These are missing in the submission.

Q. The reviewer is happy to improve the rating if the authors can report results on well-studied benchmarks in the same Habitat simulator they use (examples: CVPR 20/21 pointgoal/objectgoal navigation challenges or RxR/VLNCE challenge) and compare to the metrics reported by previously published work on these well-studied benchmarks.

-----

**[Quality] Questions about training:**
* Q. **How are the predictions of the distance-prediction module made consistent with each other?** While $\mathcal{G}_D$ predicts distances of nodes to the goal, how is the triangle inequality enforced? Eg. $\mathcal{G}_D(\cdot, \cdot, n_1) = 3$; $\text{travelcost}(n1, n2) = 2$, then $\mathcal{G}_D(\cdot, \cdot, n_2) \leq 5$.
* Q. **Could the authors include metrics for intermediate checkpoints to understand if the training of end-to-end RL baseline was close to saturation:** (L276) 20 Mn frames for PPO seems less given that 75Mn frames are needed for PPO to learn simpler PointGoal.
* Q. **Fairness of comparisons in Tab. 1 and 2**. End-to-end RL sees 10 and 20 Mn samples/steps for Tab. 1 and 2 (that too from an impoverished exploration policy). The supplement suggests 10-15 and 20-25 epochs over 1 and 2.5 Mn steps from a perfect exploration policy. Could the authors comment about the comparison between these two approaches in terms of the number of experiences seen or the number of gradient updates made?
* Q. **Another way of no simulation**: Saving graphs with each pose was done in basic image-goal navigation in [11]. This too can train agents without simulating these observations. Could the authors comment on the disadvantages of this perfect model-based approach for training?

**[Clarity] Minor comments (no impact on reviewer rating, solely for constructive feedback):**
* Should the “navigate to sub-goal” step in Algorithm 1 have $\rho_{sg}$ and $\phi_{sg}$ instead of $\rho_{i}$ and $\phi_{i}$?
* Also in Alg. 1, "if $\eta_s$ then" suggests any non-zero probability leads to termination. Is there a threshold here?
* The grammar in L217 is off.
* Epochs of training should be included in L277.
* L139: subscript could be changed to ‘t’ to match Fig. 4’s notation.
* Figure 3: Notation for N is overloaded: -- the set of all nodes and the number of nodes.
* L52: check grammar in “predicts the if the goal”.
* Fig. 2: It could be clarified that “goal location” isn’t already known to the agent.
* Tab. 4: Caption “passive video data, on Image-Goal…” → “passive video data, and tested on Image-Goal…”


**Time Spent Reviewing:**

8-10 hours

---

> ### Author Response · Authors · 2021-08-10
> **Response to D3yc**
>
>
> 1. *Originality*: Thanks for pointing out the related work. Our current literature review mostly focused on navigation. We would be happy to contextualize our work with respect to some of the work pointed out by the reviewer. We will include all papers and the discussion in the camera ready version of the paper.
>
> 2. *Connections to OfflineRL*: Thank you for raising this. Offline RL gets rid of online data gathering but still assumes access to learn a policy using rewards from observed trajectories. We do not use any rewards. In fact, we do not even try to learn a policy! Instead, our paper reduced navigation to a simple self-supervised vision problem -- given two images (Current, Goal) predict how far apart they are. This metric - when paired with a simple greedy strategy (choose the node with lowest predicted travel cost to the goal) - outperforms state of the art policy learning approaches (including online RL).  Furthermore, our self-supervised formulation has better convergence in practice than Q-learning does, since it does not require discount factors, bootstrapping, and/or other complicated credit assignment strategies. We will be happy to put this discussion in the main paper.
>
> 3. *Differences to [23]*: [23] is more aligned to offlineRL although it does not use reward data which offline RL uses. [23] is still using RL to learn a Q function to determine which action to take at each time step. NRNS instead uses a much simpler greedy policy where we select subgoals based on direct distance to goal prediction. Subgoal distance prediction is based on a topological graph of visual and spatial memory, allowing for learned propagation of semantic information via the graph network. Additionally, the NRNS approach allows for backtracking and making larger exploration decisions than a single step.
>
> 4. *Comparison to [4]*:  Our paper builds upon NTS[4] but there are three significant differences
> * NTS requires access to ground-truth maps and therefore still uses simulators. It uses simulator maps to sample images randomly and get ground-truth distances between pairs of images. We propose a completely different methodology of creating topological maps and estimating distances from just passive YouTube videos. We believe getting rid of the simulator requirement is a big step.
> * NRNS does not require panoramic observations. The Topological Map representation and Global Policy and target prediction ($\mathcal{G}_{T}$) in NRNS is different from NTS to handle non-panoramic observations.
> * But most importantly, the results in NRNS are significantly stronger with much fewer assumptions [just using YouTube videos to outperform on simulator tests even]. We believe this by itself makes it indispensable to the community.
> * Again we emphasize, NRNS is more scalable and more efficient than NTS. We can train NRNS with any video data with an arbitrary field of view. NRNS is also more effective as it can utilize all past observations to estimate distances. It is not possible to train the NTS model with video data.
>
> 5. *Comparison to [10]*:  We believe NRNS is significantly different from ANS[10]. Just to give few examples: (a) NRNS does not require access to ground-truth maps, ANS requires it to compute the reward; (b) NRNS does not require active interaction with the environment (no simulation), ANS requires simulation for learning the global policy using RL; (c)  NRNS learns a goal-oriented distance function, ANS learns a goal-agnostic exploration policy.
>
> 6. *Comparison to ‘learning from demonstrations/observations’* : We agree that our work is closest to learning from demonstration work and that is why most of our baselines are learning from demonstration baselines. The key differentiating factor is that most IL work assumes: (a) access to actions (e.g. teleop in simulator); (b) and human demonstrations are optimal. The goal is to learn policies that match those demonstrations by either mimicking the actions [if actions are available] or formulating rewards using the demonstrations. Note our approach does neither since it does not assume human trajectories are shortest-path and it does not compute any rewards as it does not require RL or credit assignment. Instead, our approach makes no such assumption and aims to learn a distance model and use greedy policy [so it could be classified as a model-based learning approach]. But what makes our approach stand out is the fact that  most IL approaches either use IL+RL or use IL data from the simulator itself. To the best of our knowledge, there has been no work that uses passive videos to show such strong results.
>
> 7. *Image Navigation Task*: Thank you for the question. Our goal is to show how NRNS can go a long way without using any RL and simulation for navigation. But what is the right task? We strongly believe to understand NRNS or any learning-based navigation approach, one should do ImageNav. Why is that? First, PointNav is NOT a semantic navigation task. It does not require learning semantics  [the bathroom is next to the bedroom so look for a mirror there] which is where most learning-based approaches make the most impact. Classical Navigation Approaches (SLAM based) which do not use RL/Simulator are already strong at PointNav tasks. Object Navigation task is definitely a possible semantic navigation task. But Object Navigation requires an object recognition module (maybe a blackbox) which needs to model object categories (what is a chair?). Overall, this muddles and confounds the understanding of the underlying navigation algorithm. On the other hand, image-based navigation is a much more grounded task and easy to explain/understand without confounding factors. We would like to point out that ImageNav has been in the field for much longer than the Habitat environment/benchmark. [11] which is one the earliest papers in recent years and started the trend to use RL for navigation also focused on this task. We will be releasing all our baselines, training/test splits. Our splits mostly build upon Habitat splits so suffer from similar biases (if any).
>
> 8. *How are the predictions of $\mathcal{G}_{D}$ made consistent with each other?*: There is no restriction on it being forced to fit the triangle inequality as the GD is predicting an estimate of getting closer to the goal. This is not the ground truth distance, it is a predicted distance, also there is also not necessarily a direct path (due to obstacles and house plans) from each node to the goal so the triangle inequality doesn't need to be enforced.
>
> 9. *Could the authors include metrics for intermediate checkpoints to understand if the training of end-to-end RL baseline was close to saturation?*: We do not believe the RL algorithm has converged at 10M steps and would improve with more training. However in the context of scalability, approaches like NRNS can also scale in performance in respect to diversity and size of training data. So, our current evaluation is at an operating point which is similar on both axes for all methods, similar compute and diversity to both approaches.
>
>    However, to highlight the fact that even with significantly more compute RL-based approaches struggle. We have improved the RL-baseline by increasing the diversity of data (sampling more initial and goal states aka increasing the number of training episodes per environment), increasing batch size and adding more layers to LSTM. Specifically, we train a Habitat implementation of DDPPO and we use 8 gpus and 12 processes per gpu allowing for a larger batch size. We also increased the number of LSTM layers from 2 to 3. Finally, we provide RL 5x more data (number of training episodes per environment) than NRNS. We report the performance at 50M (5x more compute than NRNS) and 100M steps (10x more compute).
>
>     At 50M, RL shows an average success rate of 25.98% for straight and 12.33% for Curved. At 100M, RL shows an average success rate of 29.01% for Straight and 15.7% for Curved. Even though NRNS is at both data (lower number of training episodes per environment) and compute disadvantage, it has performance of 47% success for Straight and 24% success for Curved test trajectories. As the results indicate, the performance improves but is still lower than NRNS. We believe if we train more, RL approaches are likely to further improve but with diminishing returns.
>
> 10. *Fairness of comparisons in Tab. 1 and 2. End-to-end RL sees 10 and 20 Mn samples/steps for Tab. 1 and 2. The supplement suggests 10-15 and 20-25 epochs over 1 and 2.5 Mn steps. Could the authors comment about the comparison between these two approaches in terms of the number of experiences seen or the number of gradient updates made?*:
> In terms of experience seen, the RL policy gets 10-20M frames of experience which are not fixed as the RL agent navigates the environment in an unrestricted manner. The $\mathcal{G}_{T}$ model is trained with clustered trajectory graphs created from the ~1 and 2.5m frames of fixed experience from the videos [much less experience]. Using the clustered graphs we train the network with each epoch having around 500k - 1m gradient updates. As shown above, even if we scale RL to 10x more computation (100M frames), our approach still outperforms it.
>
> 11. *Another way of no simulation: Saving graphs with each pose was done in basic image-goal navigation in [11]. This too can train agents without simulating these observations. Could the authors comment on the disadvantages of this perfect model-based approach for training?*:
> We are sorry to say even after reading the paper [11] we could not find the part of their approach which is  saving graphs with each pose. Can the reviewer provide a more explicit description of what they mean?
>
> 12. *Minor comments (no impact on reviewer rating, solely for constructive feedback)*:
> We thank the reviewer for pointing out the writing and grammar issues. We will fix these and we will take a thorough proofread of the document.

---

> > ### Comment · Reviewer_D3yc · 2021-08-25
> > **Final comments**
> >
> > 1+2. Thanks for clarifying the scope of the literature review is mostly navigational papers. Note, _visual navigation_ tasks with human demonstrations (VLN, ALFRED) include a teacher-forcing baseline i.e. imitation leaning only on pre-collected data. This requires no simulator (i.e. for interactive learning, but obviously needed to collect trajectories) or rewards, similar to NRNS.
> > * Interactive Instruction following: The neural model for ALFRED [Sridhar et al. CVPR 2019](https://arxiv.org/abs/1912.01734) is a seq2seq that requires no simulator or rewards, but a cross-entropy loss over demonstrations from agents. No simulator is instantiated during training, [reference: their code](https://github.com/askforalfred/alfred/blob/master/models/train/train_seq2seq.py).
> >
> > * Vision-and-language _navigtaion_ has a range of works testing teacher-forcing (TF). Again, this needs no interaction or rewards from the simulator. Some examples include the dataset paper [Anderson et al. CVPR 2018](https://arxiv.org/pdf/1711.07280.pdf) testing TF, the behavior cloning baseline, and Sec. 3.3 in [Tan et al. NAACL 2019](https://arxiv.org/pdf/1904.04195.pdf), and TF methods in Tab. 4 [Li et al. EMNLP 2019](https://arxiv.org/pdf/1909.02244.pdf#page=5).
> >
> > * CARLA: [Fu et al. 2019](https://arxiv.org/pdf/2004.07219.pdf) formulate offline autonomous navigation in CARLA, [Rhinehart et al. 2019 ICCV 2019](https://openaccess.thecvf.com/content_ICCV_2019/papers/Rhinehart_PRECOG_PREdiction_Conditioned_on_Goals_in_Visual_Multi-Agent_Settings_ICCV_2019_paper.pdf) consider offline planning in multi-agent settings, and [Codevilla et al. ECCV 2018](https://openaccess.thecvf.com/content_ECCV_2018/papers/Felipe_Codevilla_On_Offline_Evaluation_ECCV_2018_paper.pdf) discuss metrics for navigation in offline settings i.e. requiring no direct interaction with a simulator.
> >
> > 3. Thanks, this clarification is useful.
> >
> > 4.
> >
> > > NTS requires access to ground-truth maps and therefore still uses simulators.
> >
> > See pt. 5
> >
> > > We propose a completely different methodology of creating topological maps and estimating distances from just passive YouTube videos. We believe getting rid of the simulator requirement is a big step.
> > * Predominant analysis (i.e. except Sec. 5.2) is on videos of the simulator (L248-253), perfectly matching the viewpoint of the agent. Hence, _no simulator required for training_ might be not the best characterization of the work -- no online learning or interaction is required.
> > * Including clear distinctions to Chang et al. NeurIPS 2020 [23] beyond L77-79 (as done in your response) would be helpful for the readers, given that their experiments are all from real videos with a significant domain shift from the simulator.
> >
> > 5.  Thanks, this clarification is useful.
> >
> > > NRNS does not require access to ground-truth maps
> >
> > `NRNS also uses ground-truth occupancy map information to create the passive videos that it learns from` when compared to IL/RL methods based on interaction with the simulator. Calculating shortest paths between two arbitrary points (L250) needs access to this information.
> >
> > L250: "A set of 2-4 points are randomly selected from the environment using uniform sampling. A video is then generated of the concatenated RGBD trajectories of the `shortest path between consecutive points`. Note that the complete video trajectory is not step-wise optimal nor is the end frame of the trajectory"
> >
> > Obtaining optimal trajectories would need the same assumption as already made for NRNS simulation videos (except ablations Sec. 5.2). That is, access to the internal environment state is assumed to calculate the shortest paths between points to create the data for passive videos. The same is be needed to calculate optimal supervision.
> >
> > 6.
> >
> > > Note our approach does neither since it does not assume human trajectories are shortest-path and it does not compute any rewards as it does not require RL or credit assignment.
> >
> > See point 5 about creation for NRNS passive videos makes an _equally strong assumption as shortest-path trajectories_ (i.e. occupancy map). Moreover, several IL/RL works have investigated learning from imperfect demonstrations (eg. RL: [Gao et al. ICML 2018](https://arxiv.org/pdf/1802.05313.pdf); IL: [Wu et al. ICML 2019](https://arxiv.org/abs/1901.09387), [Cao et al. RAL 2021](https://arxiv.org/pdf/2103.05910.pdf), [Tangkaratt et al. ICML 2020](https://arxiv.org/abs/1909.06769)).
> >
> > > ... or use IL data from the simulator itself. To the best of our knowledge, there has been no work that uses passive videos to show such strong results.
> >
> > This is unclear. The passive videos are created by a policy being executed in the environment. This setting is the same as learning from demonstrations/observations where a policy (human or learned) is executed in the environment.
> >
> > 7.
> >
> > > Why is that? First, PointNav is NOT a semantic navigation task
> >
> > It is unclear why NRNS holds merits only when there are semantic priors to layouts. The reviewer thinks NRNS is, in principle, general enough to not be restricted by this.
> >
> > >  Object Navigation task is definitely a possible semantic navigation task...needs to model object categories (what is a chair?). Overall, this muddles and confounds the understanding of the underlying navigation algorithm.
> >
> > The goal _needs to be recognized or defined in some way_. Even in image goal navigation perception is still needed to signal that a particular target has been approximately reached via feature/representation learning consistent with objects in the observation.
> >
> > The response here, in the eye of the reviewer, isn't well-founded.
> >
> > > We would like to point out that ImageNav has been in the field for much longer than the Habitat environment/benchmark. [11] which is one the earliest papers in recent years and started the trend to use RL for navigation also focused on this task.
> >
> > The reviewer's comment was: "The reviewer is happy to improve the rating if the authors can report results on well-studied benchmarks in _the same Habitat simulator they use_" to refer easy-to-test but well-studied benchmarks for embodied navigation. Perhaps this was misunderstood. The reviewer understands [11] well (see pt. 11), but since this benchmark isn't well maintained and hasn't been used in this submission, the reviewer is unsure how this response is relevant?
> >
> > Overall, for pt. 7, *a new dataset has been proposed to test the impact of a new approach*. The point stands that several well-studied embodied benchmarks are suitable for NRNS. If a new dataset _has_ to be proposed, isn't this approach very restrictive that it cannot be applied to _any_ of the 10+ tasks that the community is already extensively working on (https://embodied-ai.org/). Concrete experiments to show performance vs. RL methods is necessary on `well-studied benchmarks` in addition to a newly introduced one will significantly help the community and add utility to this work.
> >
> > > Our splits mostly build upon Habitat splits so suffer from similar biases (if any).
> >
> > The paper released yet another benchmark for visual navigation for the community. In the current state, the paper doesn't provide sufficient dataset details and analysis (statistics, biases, comparison to existing benchmarks, exploration policy to collect passive videos, etc.) to release a new benchmark for our community.
> >
> > 8. Yes, $\mathcal{G}_D$ is the _predicted_ distance. Triangle inequality isn't a _restriction_ but a property that a distance measure should follow. An estimate of that would show that the predicted distance is consistent with what we _expect it to capture_.
> >
> > 9+10. Thanks for including the new results. It would help the paper to fairly compare the methods on the number of frames in env and also on the number of gradient updates.
> >
> > 11. Thanks for the question. Yes, this work isn't well maintained. In the implementation, Zhu et al. stored a state-action transition graph, where graph[i][j] is the location id of the destination by taking action j in location i, and -1 indicates collision while the agent stays in the same place.
> > (https://github.com/jkulhanek/visual-navigation-agent-pytorch). Hence, this captures a perfect model of the environment. Highlighting the pros of model-based methods over this would add to the value of the work (eg. when the graphs become intractably large etc.)
> > The graphs for AI2Thor scenes for [11] are available at: http://vision.stanford.edu/yukezhu/thor_v1_scene_dumps.zip
> >
> > 12. Happy to share constructive feedback.

---

> > > ### Author Response · Authors · 2021-08-26
> > > **Second response to D3yc**
> > >
> > > Dear Reviewer D3yc,
> > >
> > > We are a bit disappointed to hear some of these comments and believe our work is being characterized unfairly. We also believe there is some misunderstanding regarding experiments based on your comments (see (a)). We urge you to reconsider considering the arguments below.
> > >
> > > a) *NRNS needs ground truth data/simulator as well (except Sec 5.2)*:
> > >
> > > We feel you are being self-contradictory -- either NRNS needs simulation or it doesn't. The truth is NRNS *DOES NOT* need any simulator or ground-truth at training time.
> > >
> > > We believe your concern is coming from looking only at the experiments done on habitat -- where training videos are generated using ground-truth meshes (Table 1 and Table 2) -- while ignoring experiments in Sec 5.2 (Table 4). We want to highlight that we only did experiments on Habitat for apples-to-apples comparison with RL. RL approaches need simulators so they HAVE to train in Habitat and we generated the passive data there. We did not want to change the training domain as well as the approach in a single experiment since one could have claimed it's the change of domain (or different dataset) that makes it work. We believe this is the scientific way of doing experiments -- change one thing at a time. In Sec 5.1, we test changing the algorithm -- from RL to NRNS, -- while the training domain remains the same. Sec 5.2, we add the change the training domain (from simulation videos to real-world videos). We believe there is a misunderstanding about Sec 5.2. Table 4 and Sec 5.2 are not an ablation of NRNS but an application of NRNS to YouTube data. This real-world video dataset contains SLAM based pose estimation which means there is no access to ground truth maps. We show in 5.2 (Table 4) that we can still outperform RL baselines when training NRNS only on these YouTube videos and never seeing a video from the simulator.
> > >
> > > This is very similar to how self-supervised visual learning approaches show first experiments on ImageNet. They ignore ImageNet labels (even though data was collected using labels) and show their approach work. Just because they train on ImageNet, we do not say they need supervision. By doing first experiments on ImageNet, they can really compare to supervised baselines. In these papers, the last sections show the performance of the approaches on images in the wild such as Flickr100M or 1B images.
> > >
> > > We want to reiterate again -- NRNS does not need access to any simulator or ground-truth data. We encourage you to re-read 5.2 again and see for yourself.
> > >
> > > b) *Offline IL etc.:*
> > > The reviewer states in their summary that “learning from demonstrations (LfD) .. (is) entirely skipped from related work, discussion, and experiments” and also suggests to run “visual navigation tasks with human demonstrations (VLN, ALFRED) include a teacher-forcing baseline i.e. imitation leaning only on pre-collected data”.
> > >
> > > We want to highlight that we indeed recognize the work done in learning from demonstrations and two of our behavior cloning baselines are exactly from that line of work. So, the reviewer's claim that we skipped LfD is false. Lines 274-276 of the paper clearly state that these two baselines use the same form of training data so they are directly comparable. We agree that related work can have more discussion and will add to the discussion of the difference between our method and offline learning (points 2 and 6 of the rebuttal) to the paper for clarity.
> > >
> > > Your comment: “NRNS passive videos makes an equally strong assumption as shortest-path trajectories“  --> is not true for Sec 5.2. The real-world videos in Sec 5.2 are not based on shortest-path trajectories. We reiterate that unlike many prior LfD approaches, NRNS does not assume (a) access to actions (e.g. teleop in simulator); (b) and human demonstrations are optimal. And our results are significantly better than other LfD approaches.
> > >
> > > c) *Image Navigation as Testbed: We have given comprehensive arguments as to why we believe Image Navigation is the right task.*
> > >
> > > First, PointGoal navigation is not a semantic task and we believe not a good testbed for learning based approaches. Object goal navigation requires another module and layer of learning -- an object detector which learns what is an object. This would make scientific experiments harder as one would have to tease out where the gains come from. We believe strongly that Image Navigation should be the first demonstration testbed for NRNS. Learning Image representation and features is required in any visual navigation framework so not an extra module or a confounding factor.
> > >
> > > Secondly, we do not believe any community should only accept papers which have benchmark task results, but also papers which have a surprising element. Lots of good algorithms have first been developed outside benchmarks and then extended to benchmark tasks. Note that many papers published in machine learning and computer vision conferences (1) do not report results on standardized benchmarks, (2) do not report results on both ImageGoal and ObjectGoal navigation, for example [2, 4, 12, 17, 18, 20, 23]. We believe object goal navigation would indeed be the next possible application and once the NeurIPS community sees this paper, it is inevitable someone would try it. We also want to highlight that we plan to share our code and benchmark with the community with the intention that it will become a standardized benchmark for image nav.
> > >
> > > Please find the code used to generate test episodes here: https://github.com/neuripsSubmission/Paper4346/blob/bd2b3b419884d63f0cca6869800b8b497b13321c/src/data_generation/trajectory_data/generate_test_instances.py This uses the same episode sampling strategy as that of the well benchmarked point navigation task.
> > > We have already reached out to the Habitat-Challenge team, who communicated to us they are willing to add this benchmark to the Habitat webpage.

---

> > > > ### Comment · Reviewer_D3yc · 2021-08-27
> > > > **Final ratings**
> > > >
> > > > The reviewer deeply appreciates the active responses from the authors. The vigorous defense of their work in their rebuttal (including the direct prods to the reviewer) is received in a healthy manner and perfectly understandable. Instead of rebutting these prods, the reviewer will try to bring a conclusive end to this discussion (due to the timeline for preliminary decisions). Also, this paper has a high probability to be accepted, hence, the reviewer writes these final comments with the intention of making the paper (even) more usable for the community.
> > > >
> > > > Consider these suggestions in your revision:
> > > >
> > > > * The reviewer had read and has re-read Section 5.2. As per the authors (and also from we reviewers), this is what the paper is _really_ about. However, it is literally _9 lines_ and a table with _no analysis and one baseline_ from prior work. Consider improving rigor in Sec. 5.2 and leading with these results in the paper.
> > > >
> > > > * Please consider adding (at least) the baseline Value Learning from Videos (Chang et al. NeurIPS 2020) baseline to your image nav dataset results. It is the closest and recent work (to section 5.2) in the literature and hasn't been empirically compared.
> > > >
> > > > * Do consider the benchmark/dataset from the above paper (YouTube House Tours Dataset). They too scrape real estate tours from Youtube like Sec. 5.2. This will make the paper head-on comparable to existing data/works. The reviewer acknowledged the authors refused the compatibility of objectgoal navigation task, but the reviewer hopes they take the suggestion with more flexible consideration. To the reviewer, NRNS surely has more potential than currently demonstrated in this submission draft. It will make the paper stronger, contribute more to the community, and the authors will gather credits for these efforts.
> > > >
> > > > * The reviewer's intention to point out the plethora of directly related work is to highlight the amount of work already done in "no rl, no simulation" (especially, within vision papers too). The presentation of the paper can be made more grounded to these existing works and the claims presented in a more responsible manner. Our (otherwise amnestic) community would greatly benefit from this service by the authors. Kindly note that BC isn't an advanced LfD baseline -- it's the very first one. Including it isn't sufficient to claim comparison to this body of work.
> > > >
> > > > * "We want to highlight that we only did experiments on Habitat for apples-to-apples comparison with RL" this is a _solid_ point that justifies the authors' use of simulated data while claiming to not use simulation. Consider adding this in the discussion to reason why 5.1 is all about simulation data (unless already there and missed by the reviewer).
> > > >
> > > > Due to concerns on experimental rigor (missing direct baselines in Sec. 5.2, no rigorous analysis of the dataset, and no results on prior benchmarks) and weak grounding to prior work, the reviewer still wouldn't lean towards acceptance. However, the reviewer is raising their rating for specifically this clear "apples-to-apples" response, helpful responses to reviewers, and earnest intentions to improve the paper.
> > > >
> > > > Finally, the reviewer hopes the authors receive the above suggestions and prior discussion in a positive light. The reviewer thanks them for their persistent efforts.

---

> ### Comment · Reviewer_dPyV · 2021-08-21
> **Anything left unclear?**
>
> Dear Reviewer D3yc,
>
> I really liked this paper and the responses of the authors were satisfying to me.
>
> I'm curious if you feel like the authors addressed your concerns and if there are any issues left.
>
> Best,
> Reviewer dPyV

---

> > ### Comment · Reviewer_D3yc · 2021-08-25
> > **Key concerns about the work**
> >
> > Dear dPyV,
> >
> > Firstly, I sincerely appreciate your note and the encouragement to come to a more unanimous conclusion. Thanks for your service.
> >
> > I've responded about my concerns in detail to the authors below, but to summarize (albeit, somewhat crudely):
> > * offline imitation learning / learning from demonstrations (LfD) being thoroughly studied in prior work but entirely skipped from related work, discussion, and experiments.
> > * demonstration data (or 'passive videos' for NRNS) being derived from the simulator itself (except ablations in Sec. 5.2), still packaged as "no simulator/simulation"
> > * 'passive videos' needing access to occupancy information to calculate _shortest-paths_ between any two points (again, except ablations in Sec. 5.2)
> > * introduction of a new simulation dataset to validate the utility of a new visual navigation approach. This is despite the availability of numerous well-studied simulated visual navigation benchmarks.
> >
> > I am happy to hear your thoughts about these key concerns. Also, very open to accepting any misunderstanding from my end (happens to the best of us!). Finally, happy to work towards any constructive feedback that helps the authors improve their contribution to our community.
> >
> > Best,
> >
> > D3yc

---

### Author Response · Authors · 2021-08-10
**To all reviewers**

We thank the reviewers for time and feedback. Before we launch into the rebuttal, we urge both the reviewers and AC to have a second look at the paper and put the right context around it. For a moment, forget related work or how it relates to offline RL/IL or is ImageNav the right task, we will answer all those questions below.
We ask a simple question -- if we asked ACs / Rs before reading this paper, do you believe an algorithm trained on simple YouTube videos (RealEstate10K) can match or outperform a navigation algorithm trained on Habitat when tested on Habitat itself. We believe our results are highly surprising that even without using a simulator and RL we can outperform methods trained in simulation on the simulator test itself. Now when it comes to testing in the real world and on a real robot, our approach has a significant advantage since there is no domain gap and no need for sim-to-real. Even if RL-based approaches are improved or scaled to tens of billions of frames in simulation, it will still have an issue of sim-to-real transfer. Therefore, we believe our community which has wholeheartedly committed to RL+Simulators deserves to see this paper and see what are the right passive data baselines for the navigation tasks.

---

### Decision · Program_Chairs · 2021-09-27

**Decision:**

Accept (Poster)

**Comment:**

This paper presents a method for learning robot navigation policies avoiding simulations and reinforcement learning.

The paper was initially on the fence, generally appreciated by reviewers but with one of the reviewers pointing several weaknesses, which were discussed with the authors and among the reviewers. The main problems were a complete omission of a discussion of offline RL, and the general responses to the reviewers ("please forget about this") was quite hand-wavy. However, the authors did provide convincing answers to many issues, and in particular explaining that their setting is sufficiently different from RL (no reward, most importantly). Of course, this still means that these parts should have covered it in the related works section, and I strongly urge the authors to modify the paper.

It was also discussed that several baselines are missing, in particular work on topological memory, and the standard ImageGoal environment was still considered a restriction.

Issues on presentation and reproducibility remained.

However, there was a near consensus that the work is sufficiently novel and nicely executed.
I recommend acceptance.